# Closing the Energy Balance using a Canopy Heat Capacity and Storage Concept – A physically based Approach for the Land Component JSBACHv3.11

Marvin Heidkamp[1,2], Andreas Chlond[1], and Felix Ament[1,3]

[1]Max Planck Institute for Meteorology, Hamburg, Germany
[2]International Max Planck Research School on Earth System Modeling, Hamburg, Germany
[3]Meteorological Institute, CEN, University of Hamburg, Germany

*Correspondence to:* Marvin Heidkamp (marvin.heidkamp@mpimet.mpg.de)

**Abstract.** Land surface-atmosphere interaction is one of the most important characteristic for understanding the terrestrial climate system, as it determines the exchange fluxes of energy and water between the land and the overlying air mass.

In several current climate models, it is common practice to use an unphysical approach to close the surface energy balance within the uppermost soil layer with finite thickness and heat capacity. In this study, a different approach is investigated by means of a physically based estimation of the canopy heat storage ($SkIn^+$).

Therefore, in a first step, results of an offline simulation of the land component JSBACH of the MPI-ESM – constrained with atmospheric observations – are compared to energy- and water fluxes derived from eddy covariance measurements observed at the CASES-99 field experiment in Kansas where only shallow vegetation prevails. This comparison of energy and evapotranspiration fluxes with observations at the site-level provides an assessment of the model's capacity to correctly reproduce the diurnal cycle. In a further step, a global coupled land-atmosphere experiment is performed using an AMIP (Atmospheric Model Intercomparison Project) type simulation over thirty years to evaluate the regional impact of the $SkIn^+$ scheme on longer time scale, in particular, in respect to the effect of the canopy heat storage.

The results of the offline experiment show that $SkIn^+$ leads to a warming during the day and to a cooling in the night relative to the old reference scheme, thereby improving the performance in the representation of the modeled surface fluxes on diurnal time scales. In particular: nocturnal heat releases unrealistically destroying the stable boundary layer disappear and phase errors are removed. On the global scale, for regions with no or low vegetation and a pronounced diurnal cycle, the nocturnal cooling prevails due to the fact that stable conditions at night maintain the delayed response in temperature, whereas the daytime turbulent exchange amplifies it. For the tropics and boreal forests as well as high latitudes, the scheme tends to warm the system.

## 1 Introduction

The land surface plays a key role in climate modeling because it regulates a number of biogeophysical as well as biogeochemical processes (Sellers et al., 1997). The former process controls the partitioning of available energy – depending on the surface

albedo – into ground as well as into turbulent heat fluxes resulting in surface temperature changes which effect the diurnal variation of the boundary layer development governing convection and cloud formation. This energy cycle is coupled with the water cycle dividing the available precipitated water into runoff, drainage and infiltration leading to soil moisture changes which influence the evapotranspiration in turn. On the other hand the biogeochemical processes are mainly represented by the terrestrial carbon sink which is strongly coupled to the water cycle through the leaves' stomatal control of photosynthesis and transpiration.

In former times, land-atmosphere interactions were associated with low vertical scale phenomena limited to the atmospheric boundary layer without impact on larger scales or the climate system. But within the last decades, many studies and research papers have proven this assumption to be false (Pitman, 2003). The development of *Land Surface Models* (LSM) used in numerical weather prediction and in climate models started by using the so-called *bucket scheme* based on the theory that the soil composes boxes which can store limited amounts of water (Manabe, 1969). A few years later, Blackadar (1976) developed a two-layer model with a thin variable surface layer influenced by changeable radiation and a thick sluggish deeper layer changing its temperature governed by a wave equation. A pioneering work in the design of LSMs was the introduction of a multilayer soil model by Deardorff (1978) who includes a new method of predicting soil moisture content.

These improvements became especially relevant on the global scale when land-atmosphere transfer schemes (including a biosphere) were included into *General Circulation Models* (GCM). These models were investigated at the same time by Dickinson et al. (1986) and Sellers et al. (1986), who demonstrated the need to evaluate current LSM developments using observation-based data. The first systematic efforts in this direction was PILPS (*The Project for the Intercomparison of Land-Surface Parameterization Schemes*) (Henderson-Sellers et al., 1993). Here, synthetic atmospheric forcing data were used to improve the parameterization of the continental surface. The first results of experiments forced with real atmospheric boundary layer data was documented by the widely quoted work of Chen et al. (1997) comparing 23 different land surface schemes.

Two years later, the conclusions drawn by the point-based PILPS experiments have been extended to global scales in the *Global Soil Wetness Project* (GSWP) (Dirmeyer et al., 1999) requiring a processed atmospheric forcing data set. Only a year later, a new project was founded that followed the idea of combining on the one hand PILPS with its local-scale character and on the other hand GSWP that is based on a global perspective; this ongoing joint project is named *Global Land Atmosphere System Study* (GLASS). In the last decade and a half, GLASS has broadly expanded and various further projects have joined. The main goal is to improve land-surface schemes for the benefit of numerical weather prediction and climate models.

In the last years, there have been considerable applications to run JSBACH, the model used in this study, in a coupled global context on long time scales to study various biogeochemical and biogeophysical aspects such as the carbon cycle (Raddatz et al., 2007; Claussen et al., 2013), natural and anthropogenic land cover change (Pongratz et al., 2008; Reick et al., 2013), vegetation cover and land surface albedo (Brovkin et al., 2013) and atmosphere-forest interaction and feedbacks (Brovkin et al., 2009; Otto et al., 2011). In addition to these aspects, the physical components, that regulate the exchange of energy and water fluxes, have been studied (e.g., Knauer et al., 2015; Hagemann and Stacke, 2015; de Vrese and Hagemann, 2016). However, the performance of JSBACH on shorter time scales such as the diurnal cycle has not been tested in the past. An exception constitutes the study of Schulz et al. (2001) who have modified the numerical time integration scheme from a semi-implicit

scheme, that does not conserve energy, to an energy conserving implicit land-atmosphere coupling scheme. This scheme has been evaluated in so-called offline experiments using data from the Cabauw (Netherland) tower on diurnal time scales.

Despite this vast development and progress in modeling land surface processes, it is still common practice for several current climate models – JSBACH included – to use a prognostic procedure to close the surface energy balance within a soil layer of a finite heat capacity. In this study, a different approach is investigated. Following Viterbo and Beljaars (1995) we close the energy balance diagnostically (i.e. neglecting the time derivative) at an infinitesimal thin layer that is located at the surface of the vegetated land. Conveniently, the new scheme is abbreviated by *SkIn* which means on the one hand that the **S**urface is **k**ept **In**finitesimal thin and on the other hand representing a layer with a negligible vertical extent comparable with a thin *skin*.

To test the performance of the scheme, we have performed, as a first step, an offline single site experiment with the land component JSBACH of the MPI-ESM (for more information see section 2.1). In an offline experiment the LSM is decoupled from its host model but forced by observation data and evaluated against observed fluxes. Therefore, initial data, forcing data as well as verification data from the DICE project (Zheng et al., 2013) (for more information see section 2.3) have been used to compare energy- and water fluxes derived from eddy covariance measurements observed at the CASES-99 field experiment in Kansas with simulated fluxes. This first experiment shall answer the question: **Does the *SkIn* scheme improve the performance in reproducing the diurnal cycle in comparison to the old heat storage concept in case of shallow vegetation?**

As a further step, a global coupled land-atmosphere model experiment with the MPI-ESM has been performed following the so-called AMIP (*Atmospheric Model Intercomparison Project*) protocol (Gates, 1992). In this experiment the MPI-ESM (with T63 resolution, i.e. 1.9°) is run covering thirty years from 1979 to 2008 with prescribed sea surface temperatures. For this global experiment an extended approach has been applied. Following Moore and Fisch (1986), we take into account the energy storage in the canopy layer by replacing the unphysical heat storage approach in the energy balance equation with a physically based estimation of the heat storage of the canopy air space as well as of the biomass itself.

The importance of the so-called *canopy heat storage* in connection with the solution of the energy balance equation has been estimated in several experimental studies on the site-level (e.g. Jacobs et al., 2008). Meyers and Hollinger (2004) have shown that the combined energy of all different types of canopy heat storages (e.g. the energy flux for photosynthesis as well as the canopy heat storage in biomass and water content) can amount to 15 % of the net radiation even for crop sites. However, the simulated estimation of the canopy heat storage for longer time scales on global scale remained unregarded. Thus, the second experiment shall answer the question: **Does the extended *SkIn* scheme (*SkIn$^+$*) show a regional impact on longer time scales, and if so, are the current biases in near surface temperature at least partly caused by the former over-simplified parameterization of the surface energy balance?**

First, the physics of the climate model used for this study as well as its modifications regarding both above mentioned new approaches are depicted, followed by the description of the data used for the single site experiment (section 2). Afterwards, the results of both experiments are interpreted (section 3) and the most important outcomes are discussed (section 4) and summarized (section 5).

## 2 Model, data and experiments

In this section, the differences between the standard model and the modified model are analysed. After that, the data and the site for the offline experiment are described and the designs of both evaluation experiments are explained.

### 2.1 Model description

JSBACH (version 3.11) is the land component of MPI-ESM (Max Planck Institute - Earth System Model, version 6.3.03). In the past, it was embedded in ECHAM (EC following ECMWF and HAM representing Hamburg), the atmospheric component of MPI-ESM (Stevens et al., 2013). Since 2005 JSBACH is a full representation of the global soil-vegetation-atmosphere transfer system (Raddatz et al., 2007) which can also be run independently in a so-called *offline* version forced by climate data. The physical core components of the land processes (energy balance, heat transport and water budget) are adopted from

ECHAM5 (Roeckner et al., 2003) with a fully implicit land surface atmosphere coupling scheme (Schulz et al., 2001). This means that the mutual boundary conditions between the land surface and the atmosphere, in form of air temperature and specific humidity at the lowest atmospheric level, are formulated as implicit functions of the surface conditions at the new time step. The surface radiation follows a scheme which allows albedo changes of the surface below the canopy (Vamborg et al., 2011) and the soil hydrology is calculated using a five layer scheme (Hagemann and Stacke, 2015). To represent the

dynamics of land carbon uptake and release, JSBACH contains the photosynthesis and canopy radiation components from the BETHY (Biosphere Energy-Transfer Hydrology) model (Knorr, 2000), a prognostic phenology scheme and components for uptake, storage, and release of carbon from vegetation and soils (Brovkin et al., 2009). Natural landcover changes are simulated prognostically by a dynamic vegetation module which includes the representation of subgrid-scale heterogeneity of vegetation classes (Reick et al., 2013). Anthropogenic land use and land cover changes are prescribed either by maps or by forcing data

of the New Hampshire Harmonized Protocol (Hurtt et al., 2011).

    To simulate land surface and soil processes in JSBACH, the energy and water exchange within the soil is described by the diffusion equations for heat and moisture on a multi-layer vertical grid extending to a depth of 10 m. The soil is divided into five layers (Hagemann and Stacke, 2015) growing in thickness with increasing soil depth. The diffusion equation for heat

$$(\rho C)_{\text{soil}} \frac{\partial T_{\text{soil}}}{\partial t} = \frac{\partial}{\partial z} \left( \lambda_{\text{soil}} \frac{\partial T_{\text{soil}}}{\partial z} \right) \tag{1}$$

is solved numerically by the method of Richtmyer and Morton (1967). In Eq. (1) $(\rho C)_{\text{soil}}$ denotes the volumetric soil heat capacity [J/(m$^3$K)], $\lambda_{\text{soil}}$ is the soil thermal conductivity [W/(m K)] and $T_{\text{soil}}$ is the soil temperature. A zero flux boundary condition for heat is applied at the bottom of the soil and at the top of the soil the temperature of the uppermost soil layer is considered as the surface temperature. Thus, this implies that the ground heat flux is the heat exchange between the first and the second soil layer. An analogous equation, that governs the vertical diffusion for moisture, is represented by the one-

dimensional Richards-Equation, which is described in detail by Hagemann and Stacke (2015). To couple JSBACH and the atmosphere, the surface energy balance and surface water balance are solved to provide the boundary conditions for the two above-mentioned diffusion equations representing a link between the atmosphere and the underlying soil. The water balance

at the surface describes the changes in surface water caused by precipitation, evapotranspiration, snow melt, surface runoff and infiltration. Additionally, the snow budget and the interception reservoir of rain and snow is determined to close the entire water balance; a detailed description of these processes can be found in the ECHAM5 documentation (Roeckner et al., 2003).

The surface energy balance is calculated by partitioning the available net radiation $R_{\text{net}}$ into the ground heat flux $G$, the turbulent sensible heat flux $H$ and the latent heat flux $LE$, where the latter two, in turn, represent a forcing for the atmospheric component in the coupled system. In JSBACH the energy balance is closed, i.e. calculated and evaluated, within the uppermost soil layer including a heat storage term $S_{\text{soil}} = C_{\text{soil}} \partial T_{\text{sfc}} / \partial t$ corresponding to the term on the left hand side that is proportional to the time derivative of the surface temperature:

$$C_{\text{soil}} \frac{\partial T_{\text{sfc}}}{\partial t} = R_{\text{net}} + H + LE + G \tag{2}$$

Here, $C_{\text{soil}}$ corresponds to the area-specific heat capacity of the uppermost soil layer [J/(m²K)]. The surface fluxes of heat, water and momentum are defined using the bulk formulation based on the surface-layer similarity theory. These can be expressed by the so-called atmospheric resistance, which is the inverse product of wind speed and drag coefficient. The latter represents a measure of the turbulence strength determined by the roughness of the underlying surface and the influence of atmospheric stratification, which is quantified by empirical stability functions derived by Louis (1979, 1982) that depend on the Richardson number. The roughness lengths as well as the drag coefficients are assumed to be different for momentum and scalar quantities (Brutsaert, 1975). Over vegetated surface, the turbulent fluxes of heat, water and momentum are also given by the resistance law. However, an additional canopy resistance is added in the calculation of the water vapor fluxes. It depends on the *photosynthetically active radiation* and on the leaf area index. In addition, it is modified by a water stress factor depending on the soil water within the root zone. All these parameterizations include variables which in turn are functions of the surface temperature. Moreover, the surface temperature appears to the forth power in the description of the outgoing longwave part of the net radiation. Also the formulation of the latent heat flux exhibits a nonlinear temperature dependence. According to these dependencies, the energy balance equation (Eq. 2) and its alterations (Eq. (3) and Eq. (4) in the following) represent complex implicit nonlinear equations.

## 2.2 Model modifications

In the standard scheme of JSBACH the surface energy balance is closed within the uppermost soil layer of finite thickness (6.5 cm) and heat capacity. However, since the absorption of radiation takes place in the uppermost micrometers of the soil, this assumption appears not realistic. Therefore, in the *SkIn* approach a surface temperature $T_{\text{sfc}}$ that corresponds to an infinitesimal thin interface between the soil-vegetation and atmosphere is calculated. Hence, in this case the prognostic energy balance (Eq. 2), that contains a heat storage term, is changed to a diagnostic energy balance equation where the surface energy balance is closed for an infinitesimal thin surface:

$$R_{\text{net}} + H + LE + G = 0 \tag{3}$$

We note that the use of the instantaneous response temperature is not a novel approach. This so-called skin temperature has been introduced by Viterbo and Beljaars (1995) to replace the old ground-surface model of the ECMWF. This approach is also

used in other land surface models, e.g. in the community Noah land surface model (Niu et al., 2011). To solve the diagnostic energy balance (Eq. 3) explicitly, the non-linear terms – which are related to the outgoing longwave radiation described by the Stefan-Boltzmann law as well as to the temperature-dependent specific saturated humidity of the surface – have to be linearized. Here, a first order Taylor approximation has been chosen. Neglecting the heat storage term results in a loss of stability in the numerical solution because the storage term exerts a dampening effect. Therefore, the surface instantly reacts to variations of the forcing data especially to intense fluctuations in solar radiation flux densities or to wind speed variations. As a consequence, the first guess of the solution using the linearizations is insufficient and an iteration is needed to stabilize the system. For this implementation a simple Newton iteration combined with a fixed-point iteration has been used where the surface temperature of the previous time step is used as a first guess starting point. Further tests have shown that it is not sufficient to update only the outgoing longwave radiation as a part of the net radiation and the saturated specific humidity every iteration step. In addition, the *drag coefficient of heat* must be included in the iteration loop as well, since it non-linearly depends on the surface temperature. Taking into account the drag coefficient of heat into the iterative procedure exerts a negative feedback ensuring the stability of the numerical solution of the energy balance equation.

In addition, the implicit numerical scheme for the heat diffusion equation of the soil layer, which is based on the Richtmyer and Morton scheme (Richtmyer and Morton, 1967), has to be adjusted since the ground heat flux no longer describes a conductive heat transfer between the two uppermost soil layers but instead depends on the heat exchange between the uppermost soil or snow layer and the overlying canopy air mass. Therefore, the ground heat flux $G = \Lambda_{\mathrm{sfc}}(T_1 - T_{\mathrm{sfc}})$ is assumed to be proportional to the temperature difference between the surface and the uppermost soil layer $T_1$. The constant of proportionality constitutes an empirically determined factor, the so-called *heat transfer coefficient* $\Lambda_{\mathrm{sfc}}$ [W/(m²K)], which was introduced by Viterbo and Beljaars (1995) (they used the notation *skin conductivity*). For the heat transfer coefficient different values depending on the *plant functional type* (PFT) are assigned – predominantly between 10 and 40 W/(m²K). The values used for $\Lambda_{\mathrm{sfc}}$ in the present study can be found in Trigo et al. (2015).

The concept of the surface temperature characterizing an infinitesimal thin surface, in which the heat storage is completely neglected, is only valid for areas where bare soil or shallow vegetation prevails and is considered as a special case which is analysed in an offline single site experiment located in Kansas' grassy landscape (for a detailed description see section 2.3). For the global evaluation experiment, that includes forest regions such as the tropical rain forest with a dense canopy of up to 45 m height, this approach is insufficient. In this case, the change in total heat content (in short heat storage) of the canopy air, water vapor and biomass itself is not longer negligible. Therefore, the canopy heat storage $S_{\mathrm{cano}}$, which is based on a formulation given by Moore and Fisch (1986), is introduced into the energy balance equation:

$$S_{\mathrm{cano}} = LE + H + G + R_{\mathrm{net}} \tag{4}$$

It is composed of the sum of three parts

$$S_{\mathrm{cano}} = S_T + S_{\mathrm{veg}} + S_q \tag{5}$$

where $S_T$ denotes the heat storage in the canopy air space, $S_{\mathrm{veg}}$ the heat storage of biomass and $S_q$ the heat storage resulting from changes in specific humidity in the canopy layer (in short latent heat storage). The heat storage in the canopy air space

$S_T$ can be expressed as

$$S_T = C_T \frac{\partial T_{\mathrm{sfc}}}{\partial t} = c_p \rho_{\mathrm{a}} z_{\mathrm{veg}} \frac{\partial T_{\mathrm{sfc}}}{\partial t} \tag{6}$$

where $C_T$ is the area-specific heat capacity of the canopy air, $c_p = 1005\ \mathrm{J/(kg\,K)}$ the specific heat capacity of air at constant pressure, $\rho_{\mathrm{a}}$ the density of air and $z_{\mathrm{veg}}$ the vegetation height. The heat storage of biomass in the canopy layer $S_{\mathrm{veg}}$ is determined as

$$S_{\mathrm{veg}} = C_{\mathrm{veg}} \frac{\partial T_{\mathrm{sfc}}}{\partial t} = c_{\mathrm{veg}} m_{\mathrm{veg}} \frac{\partial T_{\mathrm{sfc}}}{\partial t} \tag{7}$$

where $C_{\mathrm{veg}}$ is the area-specific heat capacity of the biomass, $m_{\mathrm{veg}}$ the area specific mass of biomass and $c_{\mathrm{veg}}$ the specific heat capacity of moist biomass according to Moore and Fisch (1986). The latter is approximated by a weighted average between the specific heat capacity of dry biomass containing a temperature dependence and the specific heat capacity of water $c_{\mathrm{w}} = 4184\ \mathrm{J/(kg\,K)}$ assuming a constant water mixing ratio. For example, at a temperature of 25 °C the canopy biomass has a specific heat capacity of $c_{\mathrm{veg}} \approx 2650\ \mathrm{J/(kg\,K)}$. The area specific mass of moist biomass $m_{\mathrm{veg}}$ can be estimated as a function of the vegetation height $z_{\mathrm{veg}}$ using a linear relationship, namely $m_{\mathrm{veg}} = \rho_{\mathrm{veg}} z_{\mathrm{veg}}$, where $\rho_{\mathrm{veg}} \approx 1.67\ \frac{\mathrm{kg}}{\mathrm{m}^3}$ is the partial density of moist biomass, i.e. the mass of moist biomass per one cubic meter of air estimated using values given by Moore and Fisch (1986).

The latent heat storage $S_q$ can be calculated according to Moore and Fisch (1986) as follows:

$$S_q = L_{\mathrm{v}} \rho_{\mathrm{a}} z_{\mathrm{veg}} \frac{\partial q_{\mathrm{cas}}}{\partial t} \tag{8}$$

where $L_{\mathrm{v}} = 2.5 \cdot 10^6\ \mathrm{J/kg}$ denotes the latent heat of vaporization and $q_{\mathrm{cas}}$ the specific humidity in the canopy air space. In contrast to the heat storages of canopy air and biomass – Eq. (6) and Eq. (7) – which are expressed by means of heat capacities related to the time derivative of surface temperature, the situation is more complicated regarding the latent heat storage: Changes in specific humidity can occur independently of temperature changes. That means, considering only changes in specific humidity due to changes in surface temperature would neglect other humidity sources and sinks. Thus, a different approach to parameterize the latent heat storage is required because the current schemes does not contain a prognostic variable for the specific humidity in the canopy air space. In this approach, we take into account the heat storage resulting from changes in specific humidity of the canopy air space by defining an effective *surface specific humidity* $q_{\mathrm{sfc}}$, which is the best proxy for the specific humidity in the canopy layer that we have. It represents a nonlinear weighted average between the specific air humidity above the canopy $q_{\mathrm{air}}$ and the saturated specific humidity at the surface temperature $q_{\mathrm{sat}}(T_{\mathrm{sfc}})$, by demanding that

$$\frac{q_{\mathrm{air}} - q_{\mathrm{sfc}}}{r_{\mathrm{a}}} \overset{!}{=} LE(q_{\mathrm{air}}, q_{\mathrm{sat}}(T_{\mathrm{sfc}}), r_{\mathrm{a}}, r_{\mathrm{c}}, ...) \tag{9}$$

where $r_{\mathrm{a}}$ is the atmospheric resistance, $r_{\mathrm{c}}$ the canopy resistance and $LE$ the latent heat flux as it is calculated in the energy balance equation. This means that $q_{\mathrm{sfc}}$ is calculated to represent the effective near surface specific humidity that is required to reproduce the surface moisture fluxes due to turbulent exchange processes. In principle, the specific humidity of the boundary layer $q_{\mathrm{air}}$ could be used as a proxy of the canopy air space humidity $q_{\mathrm{cas}}$ as suggested by Moore and Fisch (1986). However,

we are of the opinion that the usage of $q_{\text{air}}$ would underestimate the latent heat storage in the current scheme. This leads to a modified formulation of the latent heat storage $S_q$:

$$S_q = L_{\text{v}} \rho_{\text{a}} z_{\text{veg}} \frac{\partial q_{\text{sfc}}}{\partial t} \tag{10}$$

Because $q_{\text{sfc}}$ is not a prognostic variable in the energy balance, its time derivative is approximated by using values of $q_{\text{sfc}}$ at previous time steps. This is an approximation that is inevitable in the current model framework and can only be avoided by developing an extended dual source canopy layer scheme which includes a prognostic specific humidity of the canopy air space as mentioned in the discussion (see section 4). Using these approaches, the former unphysical soil heat storage concept in the surface energy balance equation has been replaced by a physically based estimation of the canopy heat storage.

When discussing heat storages within the canopy one also has to consider the energy stored in form of chemical energy by carbohydrate bonds through the process of photosynthesis. Following Nobel (2009) the energy required to incorporate 1 mol $CO_2$ is 479 kJ. That means, a $CO_2$ flux of 1 mg $CO_2/(m^2 s)$ corresponds to an energy flux of about 11 $W/m^2$. This chemical heat storage has been evaluated in several experimental studies on the site-level (e.g. Jacobs et al., 2008; Meyers and Hollinger, 2004). However, in these considerations emphasis has only been on the energy consumption through photosynthesis (GPP, i.e. gross primary production). The aspect that heat will also be released during the process of plant respiration (e.g. Wohl and James, 1942; Thornley, 1971) has been neglected in most studies. This had led to an overestimation of the chemical heat storage. Thus, one has to consider not only the net primary production (NPP) but also all other processes that release $CO_2$. Therefore, we added an additional term in the energy balance to estimate the magnitude of the heat stored in chemical bonds:

$$S_{\text{cano}} = LE + H + G + R_{\text{net}} + \beta F_{CO_2} \tag{11}$$

where $F_{CO_2}$ is the net $CO_2$ flux in $kg/(m^2 s)$ and $\beta = 10.884 \cdot 10^6$ J/kg the above mentioned conversion factor. The net $CO_2$ flux $F_{CO_2}$ is calculated in JSBACH using the photosynthesis scheme of Farquhar et al. (1980) for C3 plants and the scheme of Collatz et al. (1992) for C4 plants. We note that the estimation of the chemical heat storage in our study is a first attempt to address this issue and should be investigated in more detail in future studies.

## 2.3 Data and site description

To address the first scientific question of this study, i.e. whether the heat storage concept correctly reproduces the coupling between the land and the atmosphere throughout the diurnal cycle in case of shallow vegetation, an offline single site simulation with the land surface model JSBACH has been performed. We use observations from the *Diurnal Land/Atmosphere Coupling Experiment* (DICE, http://appconv.metoffice.com/dice/dice.html). This experiment was a joint effort between GLASS and GEWEX (*Global Energy and Water Exchanges*). The goal of DICE was to identify the complex interactions and feedbacks between the land and the atmospheric boundary layer. Koster et al. (2006) identified so-called *hot spot* regions characterized by a high coupling strength between land surface and atmosphere, which means the degree to which anomalies in land surface variables, for example soil moisture, can affect the generation of precipitation or other atmospheric processes. Moreover, there has been a disagreement among models for these regions in the past. One of these *hot spot* regions is located in the great central

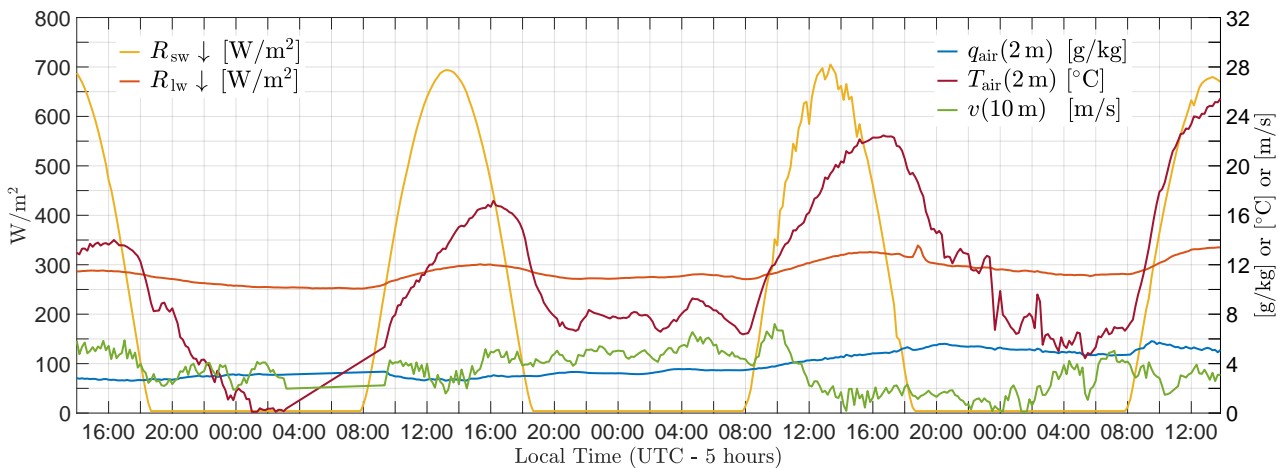

**Figure 1. DICE forcing data used for the offline single-site experiment** in form of longwave downward radiation $R_{lw}\downarrow$ (orange) and shortwave downward radiation $R_{sw}\downarrow$ (yellow) in W/m$^2$ (see left axis), as well as 10 m wind speed $v$ (m/s, green), 2 m air temperature $T_{air}$ (°C, red) and 2 m specific humidity $q_{air}$ (g/kg, blue) (see right axis). Data are from the CASES-99 Experiment in Kansas from October 23rd 1999 to October 26th 1999.

plain of the United States. Therefore, DICE uses data from the CASES-99 (*Cooperative Atmosphere-Surface Exchange Study - 1999*) field experiment in Kansas (37.7 N, 263.2 E). DICE principally follows the concept described in Steeneveld et al. (2006) and Svensson et al. (2011) regarding the same three days from the afternoon (19 UTC, 2pm local time) of October 23rd 1999 to the 26th. For these three days, DICE provides forcing data (precipitation, air pressure, air temperature, specific humidity and wind, as well as short- and longwave incoming radiation) and verification data (surface temperature as well as sensible and latent heat fluxes) with a high temporal resolution of 10 minutes (Fig. 1). In addition, 10-years forcing data of lower resolution (3 hours) are available for an initialization.

The measurement site was located near Leon representing a relatively flat homogeneous terrain with dry soils. Being far off from the ocean or large bodies of water it is dominated by a continental climate. Following *Köppen climate classification* it belongs to the northern limits of North America's *humid subtropical climate* zone (*Cfa*). Its climate is characterized by hot, humid summers and cold, dry winters. Without any major moderating influences such as mountains there are often extreme weather events such as thunderstorms or tornados in the spring and summer months. Over the course of a year, the average annual precipitation is, with 993 mm distributed over 147 rain days, comparably high, because convective precipitation prevails over stratiform or orographic precipitation meaning the rain events are rather severe and short-lasting then weak and long-lasting.

The actual experiment of DICE contains the above-mentioned three days from the afternoon of October 23rd 1999 to the 26th. The three days are part of a 25 days lasting drought and characterized by an increasing trend in temperature without any precipitation and permanent clear skies. The value of the air temperature of the first day and particularly its night is below the

October's average whereas the second night is relatively warm (Fig. 1). These different conditions during the nights indicate various turbulence and atmospheric stability regimes: *intermittent turbulence* (transition from lightly unstable to lightly stable conditions) for the first night, *continuous turbulence* or *fully turbulent* (neutral, tendency to lightly stable) for the second night with high wind speeds and *radiative* (hardly any turbulence and very stable) for the third night including a temporary calm.

### 2.4 Design of evaluation experiments

For the first offline single site experiment an almost ten year spin-up is run to ensure an equilibrium temperature and moisture in deeper soil layers. This initialization is done using forcing data of the Water and Change (WATCH) project (Weedon et al., 2014), which bases on the forty years ECMWF Re-Analysis (ERA-40) data. The spin-up's last year is replaced by a local measurement site in Smileyberg, Kansas, ending with the first day of the actual three-day experiment. Gaps of this last year are filled by values from the WATCH data, so that the time series contains no missing values. In summary, the spin-up data contains 3583 days with a time step of three hours, which was interpolated to an hourly model time step. The actual three-days simulation is performed with a model time step of 10 minutes. The surface and soil parameters of the model (root depth, roughness length, etc.) were adjusted to the site's properties.

The second evaluation experiment is run in a global coupled model configuration for thirty years from 1979 to 2008 with a T63 resolution (i.e. 1.9°). The simulation follows the AMIP project (Gates, 1992), which means that the sea surface temperature is prescribed. The soil and surface parameters of the model are the standard values, which can be found in Hagemann (2002), and the time step of the model is 450 seconds. Data from the WATCH project (Weedon et al., 2014) are used to compare the model results with observations.

## 3 Results

In this section, first, the results of *SkIn* are evaluated in an offline single site experiment located in Kansas, where shallow vegetation prevails and the canopy heat storage is negligible. After that, the extended scheme *SkIn*$^+$ including the effect of the canopy heat storage is discussed in form of a global experiment.

### 3.1 Single site experiment

Figure 2 shows time series of various quantities in the surface energy balance equation for the three specific days of the DICE experiment. Plotted are calculated fluxes of net radiation, sensible heat, latent heat and ground heat flux using the standard version of JSBACH (upper panel), as well as the modified version *SkIn* (lower panel). In addition, observational data (dashed lines), which are considered as verification data, are also plotted. Considering first the typical behavior of the observed diurnal cycle, the energy in form of net radiation (violet) is divided into the sensible (red), the latent (blue) and the ground heat flux (green). With respect to the sign convention of the fluxes, we note that negative (positive) turbulent fluxes are pointing downwards (upwards) and are related to an uptake (release) of surface energy. Positive (negative) ground heat fluxes constitute an energy gain (loss). Heat fluxes measured by the eddy covariance method usually do not close the energy balance, that

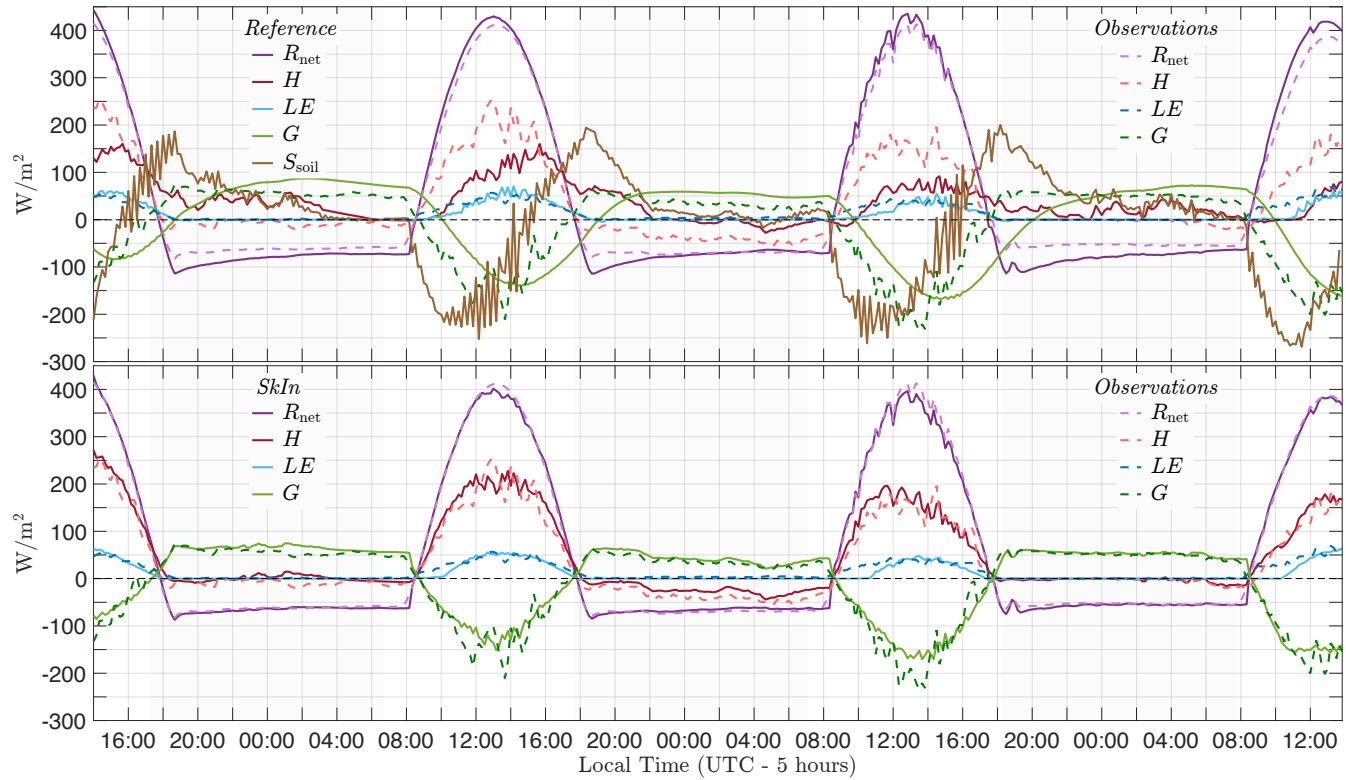

**Figure 2. Performance of the JSBACH scheme on diurnal time scales:** Comparison of time series of the components of the surface energy balance equation between the reference model (upper panel) and *SkIn* (bottom panel) against observations (dashed lines). Plotted are net radiation $R_{\text{net}}$ (violet), sensible heat flux $H$ (red), latent heat flux $LE$ (blue), ground heat flux $G$ (green) and heat storage term $S_{\text{soil}}$ (brown). Data are from the CASES-99 Experiment in Kansas from October 23rd 1999 to October 26th 1999.

means they are smaller than the available energy (net radiation minus ground heat flux, e.g. Twine et al., 2000). Normally, this imbalance is distributed to the heat fluxes weighted by the Bowen ration (see e.g. Ingwersen et al., 2015). However, since the ground heat flux was not measured during the DICE experiment, there is no other possibility than to calculate it as the residuum including the stated imbalance. Therefore, the ground heat flux should only be used as an approximation, especially during the
5 day where the largest unclosure can occur.

At daytime, the net radiation is positive with a maximum when the sun is at its zenith, whereas at night it stays at a constant negative value which results in a heat loss of the soil corresponding to a positive ground heat flux pointing upward. During the first and third night, the sensible and latent heat disappear because turbulent motions are suppressed under stable conditions, whereas in the second night a negative sensible heat flux prevails – meaning the atmosphere releases heat to the soil.
10 The latent heat flux reaches merely $50\,\text{W/m}^2$ during these three days, which is the result of the 25 days lasting drought. Thus, with about $250\,\text{W/m}^2$, the sensible heat flux represents a large part of the available energy (about $400\,\text{W/m}^2$) leading to a high Bowen ratio of about 5:1. Regarding the reference run, the sensible, the latent and the ground heat flux react slower to the

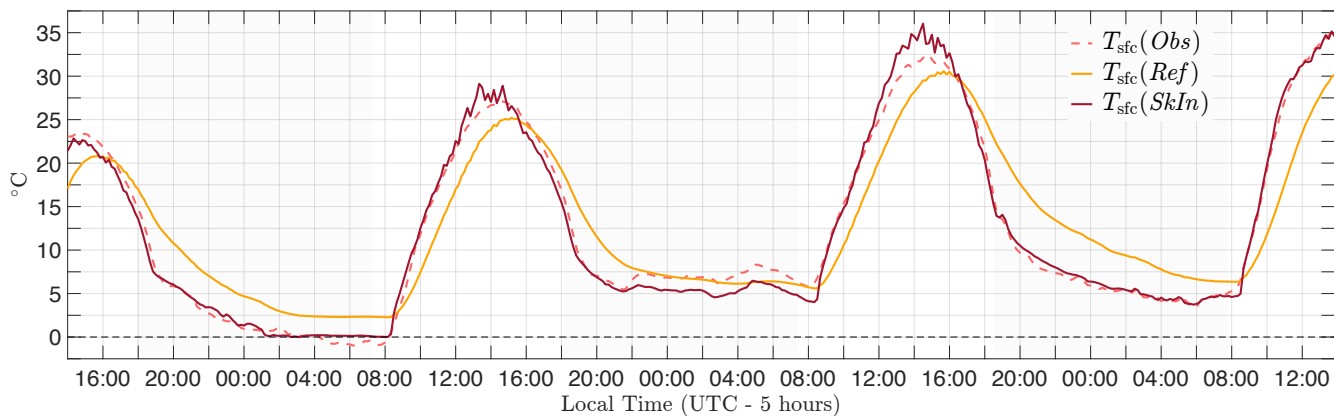

**Figure 3. Performance of the JSBACH scheme on diurnal time scales:** Comparison of time series of the surface temperature $T_{\text{sfc}}$ between the reference model (orange line) and *SkIn* (red line) against observations (dashed line). Data are from the CASES-99 Experiment in Kansas from October 23rd 1999 to October 26th 1999.

increase in net radiation. The cause of this delay is the presence of the thermal energy storage $S_{\text{soil}}$ within the uppermost soil layer which amounts to $250\ \text{W/m}^2$. This energy is stored (negative flux) during the day and released (positive flux) during the night. Therefore, the assumption that the energy will be absorbed in a layer of soil of $6.5\ \text{cm}$ thickness results in a phase shift of about two to four hours. In nature, radiation is absorbed within the first few micrometers of the soil-vegetation system and is
then transported via thermal conduction further downwards. As a consequence of the change of the heat storage, the uppermost soil layer is heated up during the first part of the day and releases a part of its energy content during the second half: As soon as the net radiation starts to increase, the heat is instantly stored in the uppermost soil layer resulting in an absolute maximum of the thermal energy change up to $250\ \text{W/m}^2$. After a delay of about two hours, this energy is partly released by the sensible heat flux and partly conducted into deeper layers by the ground heat flux. Thus, the uppermost soil layer continuously absorbs
less energy until around 4 pm (local time), when the situation is reversed and the layer releases the accumulated amount of energy it has absorbed previously. In some cases, a part of this energy transfer persists until night resulting in nocturnal heat releases that destroy the stable boundary layer. A further weakness of the reference scheme is related to its susceptibility to amplify fluctuations. This can be seen for example in the time series of the heat storage term jumping from one time step to another by about $150\ \text{W/m}^2$.
Comparing the results of the modified *SkIn* model version with those of the reference run, we note that the first improvement is the disappearance of the nightly heat releases. The sensible heat flux of *SkIn* follows the observations almost perfectly; even in the second night where negative heat fluxes occur. In the *SkIn* simulation, where per definition no heat storage exists, we find that all fluxes immediately react to variations in the radiative forcing and the phase-shift found in the reference simulation vanishes.
The surface temperature exhibits a similar phase-shift (Fig. 3): in the reference simulation the surface temperature is under-estimated by up to $4\ \text{K}$ in case of heating and overestimated up to $8\ \text{K}$ in case of cooling with respect to the observations. The

simulation of *SkIn* shows only some minor disagreement with the observations. In particular, *SkIn* overestimates the surface temperature maximum in the second and the third day but fits the observation apart from that quite well. The behavior of the surface temperature in the reference run exhibits a phase shift as it is equal to the soil temperature at about 3 cm depth. Here, the ground heat flux, as the heat exchange between the first and second soil layer of the reference run, shows the same inertial lagging (Fig. 2). In addition, it is quite smooth and overestimates the nightly ground heat flux (particularly in the first night). The same phase shift in temperature as well as the delayed response in the heat fluxes has also been found by Betts et al. (1993).

Interestingly, the phase error with respect to observations in the temporal course of the surface temperature caused by the dampening effect exerted by the heat storage exhibits an asymmetric behavior. The phase-shift between the surface temperature and the observed temperature increases with time and is much larger during the night than during the day. In the *SkIn* scheme, the adjustment to an equilibrium temperature, which is determined by the radiative forcing, is achieved instantaneously, whereas in the approach that includes a heat storage the temperature difference between the simulated temperature and the equilibrium temperature decreases over time according to an exponential rate. The time required to reach the equilibrium state is determined by a time constant which depends on the turbulence conditions in the atmosphere. During the day the turbulent motions intensify the turbulent exchange and reduce the time to reach the equilibrium. In contrast, at night the exchange is strongly reduced under stable conditions resulting in larger relaxation times. As a result, the simulated temperature in the *SkIn* run is always lower than that in the reference run in the afternoon and during the night. Moreover, the skin conductivity like the drag coefficient – in contrast to the heat capacity – acts to reduce the relaxation time to reach the equilibrium. However, we agree that the incorporation of a skin conductivity as well as the drag coefficient also damps the amplitude of the response in surface temperature to variations in the forcing. Overall, the conclusion can be drawn that on the basis of a daily average the cooling effect of *SkIn* outweighs its small warming effect during the day for regions where shallow vegetation prevails (here *SkIn* leads to a cooling of 0.6 K).

In the next section, this finding will be examined further using an AMIP experiment. Moreover, we will address the question to what extent the surface processes of regions with tall vegetation or located in high latitudes without a pronounced diurnal cycle will respond to the formulation of the land-atmosphere coupling.

## 3.2 Results of the AMIP run

A key aspect of the $SkIn^+$ scheme is the introduction of the canopy heat storage $S_{\mathrm{cano}}$. Since the latent heat storage $S_q$ cannot be completely related to a temperature tendency, it is not possible to compare the heat capacities related to different processes, but one has to compare different heat storages. Because heat storages have the nature to compensate each other over longer time scales, we compare only positive contributions of the heat storages to estimate their magnitude. This could be interpreted as the average amount of energy that is stored in the canopy. The same amount will also be released. The thirty year mean of the canopy heat storage ranges between roughly 5 and 15 W/m$^2$ for high vegetation (Fig. 4, upper panel). The total canopy heat storage $S_{\mathrm{cano}}$ amounts to 3.7 W/m$^2$ in the global land mean. This value appears relatively small because of the fact that the regions with no or shallow vegetation account for negligible storages around 0 W/m$^2$ and do not contribute to the mean. The heat storage in the canopy air space $S_T$ amounts to 19 % of the total canopy heat storage and the latent heat storage $S_q$

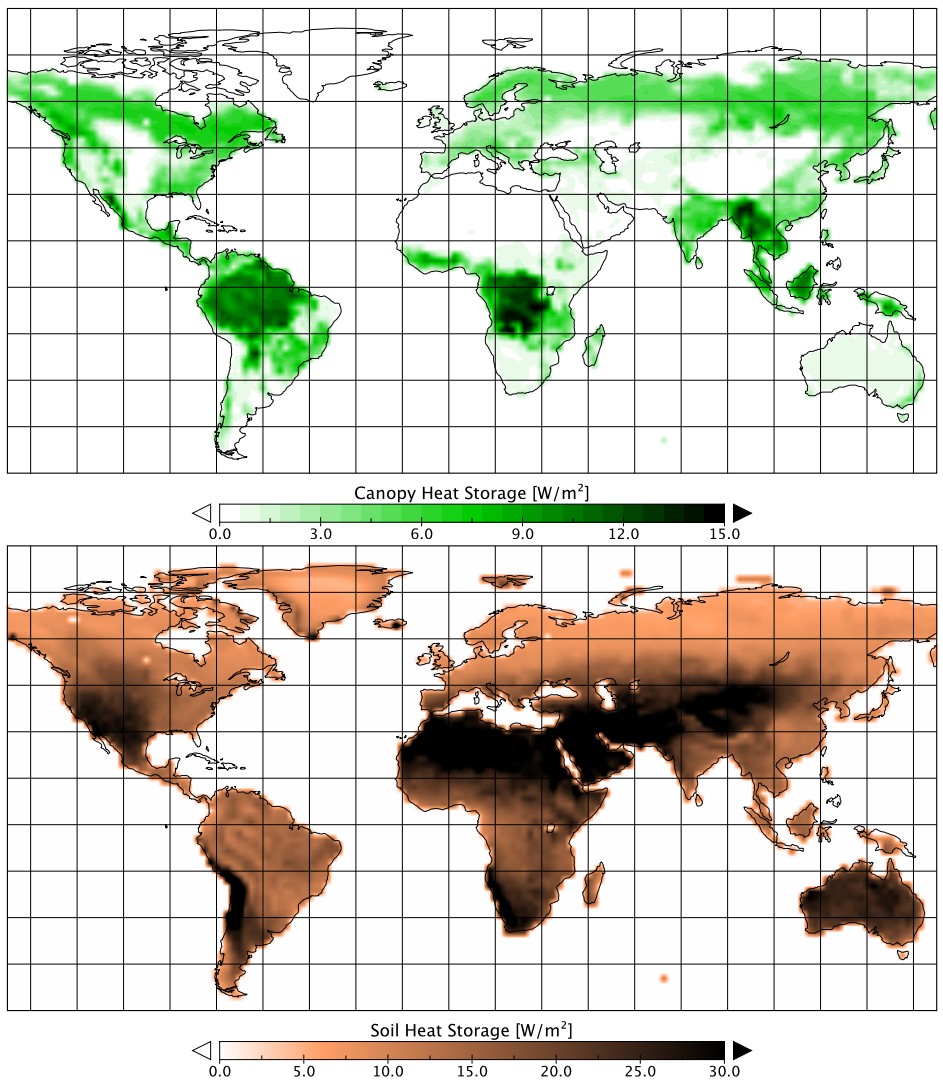

**Figure 4. Comparison of Canopy and Soil Heat Storage:** Global distribution of the positive contributions of the Canopy Heat Storage $S_{\mathrm{cano}}$ (upper panel) and the Soil Heat Storage $S_{\mathrm{soil}}$ (bottom panel), both in W/m$^2$ as a thirty-year mean (1979-2008).

constitutes 22 %. The most significant contribution to the total storage with around 60 % is given by the heat storage of the moist biomass $S_{\mathrm{veg}}$. In the warm and humid tropics with high vegetation prevailing the largest values for the canopy heat storage are found. Here, the latent heat storage partly shows a similar magnitude as the biomass heat storage. In the tropics the total canopy heat storage averages to 12 W/m$^2$ (15 W/m$^2$ at maximum), whereas in the taiga mean values of 5 W/m$^2$ appear, 5 and in deciduous forests mean values of 3 W/m$^2$ are found.

Comparing the latent heat storage $S_q$ qualitatively on diurnal scales with the other two storage terms, which are directly related to the surface temperature tendency, we find that the temperature related heat storages tend to react like a common heat

storage, which exhibits a positive peak during the first half of the day and a negative during the second part (compare the soil heat storage in Fig. 2), whereas the latent heat storage does not show this temporal course. It shows positive as well as negative changes in heat storage during the whole daytime. This corresponds to the fact that the specific humidity does not follow a strict diurnal pattern as the surface temperature. On the contrary, there are different kind of days representing either a positive

or negative trend in humidity depending on dry or wet periods.

The positive part of the chemical heat storage $\beta F_{CO2}$ follows the same regional pattern as the other canopy heat storages whereas the temporal course on diurnal scales differs. In particular, the chemical heat storage follows the PAR (photo active radiation) part of the incoming solar radiation and results in an energy consumption during the day and an energy release due to respiration in the night. With 0.64 W/m$^2$ on global average it amounts to 17 % of the total canopy heat storage $S_{\mathrm{cano}}$

and is slightly smaller than $S_T$. Nonetheless, it should not be neglected because the sum of all these supposedly small terms could be important. As mentioned before, we think that the chemical heat storage is an interesting aspect which should also be investigated in the connection of the interaction between the carbon cycle and the climate on longer timescales.

The soil heat storage in the reference model (Fig. 4, bottom panel) is related to the soil heat capacity of the uppermost soil layer, which is determined by the present soil type based on the FAO soil classifications. The soil heat storage varies spatially

in the range between 5 and 50 W/m$^2$ and amounts to 17 W/m$^2$ on global average. For regions with tall vegetation it reaches values of about 10 W/m$^2$, which corresponds to the same order of magnitude as the canopy heat storage in the $SkIn^+$ scheme: for tropical forest it is slightly smaller; for northern hemisphere forests it is slightly larger. However, the soil heat storage significantly exceeds the canopy heat storage in regions with no or low vegetation. In general, the magnitude of the canopy heat storage as well as the soil heat storage is proportional to the temperature tendency. The regions with no or low vegetation

exhibit the largest diurnal range in temperature. Therefore, the largest discrepancy appears here and amounts up to 50 W/m$^2$. Thus, we expect that the main influence of the $SkIn^+$ scheme occurs in regions where bare soil or shallow vegetated regions, such as grass lands or the savanna, dominate, while we expect a rather small effect in forest regions.

Figure 5 illustrates the performance of the $SkIn^+$ scheme, which includes the canopy heat storage $S_{\mathrm{cano}}$, on regional scales using a thirty-year average for the summer half year (April to September) (upper panel) by displaying the difference of the near

surface temperature between $SkIn^+$ and the reference run. Based on our experiences with the offline version, we know that $SkIn$ leads to a warming during the day and to a cooling in the night because of its instantaneous response to the radiative forcing. Thus, the sign of the local mean temperature difference between $SkIn^+$ and the reference run depends on the fact whether the night effect prevails or whether the daytime effect and other processes predominate such as clouds and precipitation. In the global mean, $SkIn^+$ leads to a cooling of 0.22 K. Almost all regions characterized by no or low vegetation together with

a pronounced diurnal cycle, where mostly well mixed conditions during the day and very stable conditions during the night occur, show an overall cooling in $SkIn^+$ relative to the reference scheme (with a maximum up to -3.5 K). This effect is clearly visible in Australia, the South West United States, the Gran Chaco region in South America, the Sahara, the Arabian region and Central Asia.

In the tropics the $SkIn^+$ and the reference scheme show much smaller differences which suggests that the canopy heat

storage in $SkIn^+$ roughly corresponds to the one of the uppermost soil layer. Only in some parts of the tropics $SkIn^+$ is slightly

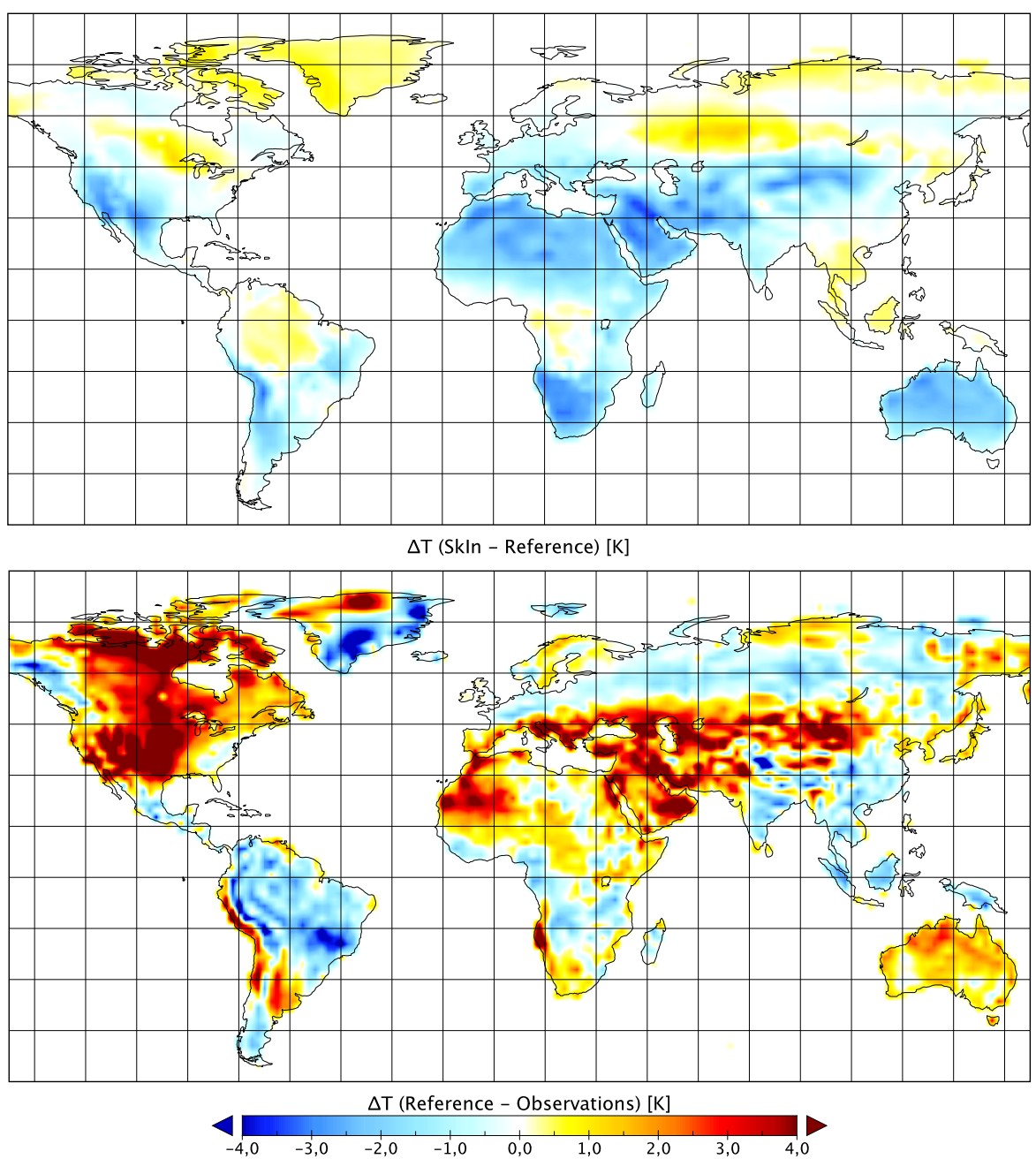

ΔT (SkIn − Reference) [K]

ΔT (Reference − Observations) [K]

−4,0    −3,0    −2,0    −1,0    0,0    1,0    2,0    3,0    4,0

**Figure 5. Performance of** *SkIn*[+] **scheme on regional scales:** Thirty-year (1979-2008) summer half year (Apr-Sep) average of the difference in near surface temperatures between *SkIn*[+] and the reference run (upper panel) as well as between reference run and observations (in short model bias, bottom panel).

warmer than the reference scheme indicating an opposite *SkIn* effect with higher temperatures at night and lower during the day. Consequently, an absence of the canopy heat storage would lead to a slight cooling in the tropics. With respect to the mid and high latitudes of the northern hemisphere we note that north of fifty degrees *SkIn*$^+$ leads to a warming in summer relative to the reference scheme because the daytime effect prevails in this region and is caused by the supply of heat during the longer insolation period in these regions during northern hemisphere summer.

Figure 5 (bottom panel) depicts the difference in near surface temperatures between the reference run and observations (WATCH dataset). A comparison of the patterns of upper and bottom panel in Fig. 5 shows that for certain regions, where the reference model tends to be too warm, *SkIn*$^+$ produces a cooling and vice versa. Not all biases disappear entirely, especially as the existing biases are much larger than the effects due to *SkIn*$^+$, but it improves the overall performance of the land surface exchange significantly by reducing the model bias. Thus, the root mean square error of the global average temperature bias over land is reduced by 0.19 K, which corresponds to a bias reduction of about 9 %. *SkIn*$^+$ leads to significant improvements in the South West United States, in the Gran Chaco region, West and Central Africa and particularly in the Arabian region, Central Asia and Australia. In some other regions, such as parts of South Africa or in North Americas boreal forests, the *SkIn*$^+$ scheme seems to be unable to reproduce the temperature patterns. Therefore, further refinements are required to improve the treatment of various land-atmosphere interaction processes, in particular over boreal forests and in snow covered regions. Moreover, also other biases, that are not related to land processes, for example caused by the atmosphere and its large-scale circulation patterns, may be responsible for the apparent short comings of the *SkIn*$^+$ scheme.

# 4    Discussion

In this study we demonstrate that the soil heat storage approach appears to be too simple and is not justified to correctly reproduce the coupling between land surface and atmosphere with respect to the simulation of diurnal cycles of energy fluxes and the near surface temperature in regions with low vegetation. *SkIn*$^+$ does not show an unambiguous effect in one direction but causes both a cooling as well as a warming depending on the time of day. It is debatable whether the heat storage approach just induces phase errors only in the diurnal cycle of surface fluxes and of near surface temperatures producing errors that cancel each other when averaged over longer time scales. This assumption, however, appears not to be true because a temperature signal of up to 3.5 K is found in the thirty-year temperature average differences. Moreover, the calculation of the correct timing of heat fluxes is an important issue per se because it influences and triggers convection that governs the formation of clouds and precipitation which in turn affects the energy fluxes. Therefore, we recommend that the *SkIn*$^+$ scheme should be used not only for models that operate on short time scales but also for Earth system models with longer time scales.

However, in some regions the *SkIn*$^+$ scheme shows a worse performance than the old scheme, likely because some existing biases only emerge in the *SkIn*$^+$ scheme. In addition, we think that the *SkIn*$^+$ scheme, which considers the canopy heat storage, would take full effect in the case where subgrid scale surface temperatures variations in a grid cell are taken into account. At the moment, the surface energy balance is solved for the whole grid box using the parameter averaging method implying that the identical surface temperature is assigned to the whole grid cell. A more promising approach that would be more suitable for the

$SkIn^+$ scheme and that allows a better representation of spatial subgrid-scale heterogeneity would be a flux aggregation method (Best et al., 2004; de Vrese and Hagemann, 2016) as it is used for example in the Tiled ECMWF Scheme for Surface Exchanges over Land model (TESSEL, Balsamo et al., 2009). Moreover, future developments of land surface exchange schemes should also take into account the vertical discretization of the thermal structure within the canopy layer which is important in case of high vegetation. Here, the temperature of the tree crown, the surface temperature under the trees, the ambient air space temperature within the canopy as well as the leaf temperature itself are differentiated (e.g., Vidale and Stöckli, 2005). The development of the $SkIn^+$ scheme is only the first step to decouple the surface energy balance from the soil layer and we think that future studies, taking into account more processes within the canopy layer to address the role of the leaf temperature and its relation to the evapotranspiration within the forest, will be capable of improving our understanding of land-atmosphere exchange processes.

## 5   Conclusions

In several current climate models it is common practice to use a prognostic procedure to close the surface energy balance within the uppermost soil layer of finite thickness and heat capacity. In this study, a different approach is investigated by closing the energy balance diagnostically at an infinitesimal thin surface layer ($SkIn$). We address the question of whether the classic heat storage concept correctly reproduces the coupling between the land and the atmosphere throughout the diurnal cycle regarding shallow vegetation. For this, we performed an offline site experiment with JSBACH, the land component of the MPI-ESM, using observations from the CASES-99 field experiment in Kansas. Analysing the surface energy balance in both schemes, we find that:

- The heat storage in the standard scheme causes a dampening effect resulting in phase errors with respect to the time-dependent behavior of the heat fluxes and surface temperatures.

- A part of the stored energy is released during the night which unrealistically destroys the stable boundary layer.

- The surface temperature simulated with the reference scheme is underestimated in case of heating during the day and overestimated in case of cooling at night.

Here, we conclude that the $SkIn$ scheme leads to significant improvements in the representation of exchange processes removing almost all biases.

In a second step we investigated the effect of the $SkIn$ scheme on longer time and larger spatial scales. The question we addressed is whether the $SkIn$ scheme shows a regional impact on longer time scales, and if so, whether the current biases in near surface temperature are at least partly caused by the former over-simplified parameterization of the surface energy balance. To answer these questions, a global coupled land-atmosphere experiment covering the years from 1979 to 2008 (AMIP run) with prescribed sea surface temperature was performed. For this global run, the standard heat storage concept is replaced by a physically motivated approach describing the heat storage of the canopy layer in the surface energy balance ($SkIn^+$). In this method, not only the heat storage of the biomass itself is taken into account, but also the heat storage of the air and its humidity

within the canopy layer. In addition, we wanted to determine whether the daily warming or the nightly cooling, which occurs in the offline site level version of *SkIn*, also prevails in the coupled run and if so in which regions which effect dominates. Comparing the simulated summer near surface temperatures of the $SkIn^+$ scheme with those of the reference run as well as to WATCH data we find that:

- The heat storage of the canopy layer must be taken into account in regions with tall vegetation (especially in the tropics). Here, the heat storage of the canopy layer is larger than that of the uppermost soil layer.

- The turbulent exchange during daytime counteracts the delayed response in near surface temperature whereas during stable conditions at night a significant phase-shift occurs.

- For most regions – especially those with no or low vegetation and a pronounced diurnal cycle – the night effect of $SkIn^+$
prevails leading to a cooling in the near surface temperature relative to the standard scheme.

- For the tropics, where the heat storage of the canopy layer is larger than the one of the uppermost soil layer the $SkIn^+$ scheme leads to a slight warming.

- For high latitudes $SkIn^+$ tends to warm the near surface air temperature due to the extended day length in the northern hemisphere in summer.

In summary, the $SkIn^+$ scheme also shows a significant global effect on longer time scales and a reduction of the model bias in several regions.

*Code and data availability.* Access to the model source code (MPI-ESM version 6.3.03, JSBACH version 3.11, SVN Revision 9050) is provided through a licensing procedure (http://www.mpimet.mpg.de/en/science/models/license/).

Data of the site experiment (DICE, http://appconv.metoffice.com/dice/dice.html) as well as the verification data of the global run (WATCH, http://www.eu-watch.org/watermip/use-of-WATCH-forcing-data) can be found online.

**Supplementary material related to this article is available online at https://www.geosci-model-dev-discuss.net/gmd-2018-17/.**

*Acknowledgements.* We would like to acknowledge the National Center for Atmospheric Research (NCAR) and all people involved in the
CASES-99 experiment. We also thank the United Kingdom's national weather service *Met Office*, particularly Martin Best and Adrian Lock, for reviving and extending this project as well as for providing all data of Diurnal land/atmosphere coupling experiment (DICE) as open access. Finally, we are very grateful for critical and helpful reviews of this paper by Dr. Joachim Ingwersen (Institute of Soil Science and Land Evaluation, University of Hohenheim, Germany) and two anonymous reviewers.

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
