# Peer review of "Closing the Energy Balance using a Canopy Heat Capacity and Storage Concept – A physically based Approach for the Land Component JSBACHv3.11"

_Geoscientific Model Development, 2018_

## Referee Comment (RC1) · Anonymous Referee #1 · 15 May 2018

In modelling the land surface one has to deal with the coupling of the atmosphere-soil interface to the atmospheric conditions, and with the coupling to the soil. This paper describes an important improvement of the JSBACH land component (JSBACH) of the MPI-EMS model. In the original (reference) scheme the temperature of the first soil layer serves both goals, coupling to the atmosphere and coupling to the deeper soil. The improvement is performed in two steps. First a skin-layer is introduced which is coupled to the first soil layer through a skin conductivity. This is assumed to work for bare soil and low vegetation. A second step is taken for tall vegetation where the thermal heat capacity is now reintroduced for the biomass influence. It is shown that this improved model better resolves the diurnal cycle of temperature. Further it is shown that it also has an impact on the average temperature, as temperature responses differ for night and day among the new scheme and the reference scheme.

The paper is well written, the subject of the paper is relevant as a correct representation of land processes is at the heart of good weather, climate and transport models. The model changes are clearly layed-out, and relevant analysis are performed to show the impact of the model changes. However, there are some issues in the paper that has to be resolved before the paper can be accepted.

Main criticism:

1) As one of the proposed changes in the surface model (SkIn) are based on ideas that has already been developed by others 20 years ago. See Viterbo and Beljaars, (1995) and the references to Betts et al. (1993) and Beljaars and Betts (1993), two points arise:

a) It would be interesting to learn why these changes has not been introduced earlier in the JSBACH scheme. Were there, for example, other positive aspects in the performance of the JSBACH model that favored a conservative approach?

b) It would be informative for the reader to compare the results/improvements found by the authors for the current model, with the findings of the authors mentioned above for the ECMWF-model.

2) The rationale for Eq. 6 is unclear. A reference to Moore and Fisch (1986) is given, but I fail to see that their approach correspond to the approach given in the current paper. It is a change in q (specific humidity) that induces a change in latent heat storage in the vegetation air column. A change in q can occur while T_sfc stays constant. Thus Eq. 6 seems not to capture the process the authors try to describe.

3) P11 L5-15: In principle I can follow the reasoning of the authors here, but I think

the situation is a bit more complicated. Indeed T_sfc in the SkIn scheme responds instantaneously to the radiative forcing, but the coupling to the soil through the skin conductivity is also present. This may also induce time (phase) shifts. Please comment.

Minor issues:

4) For clarity it is good to mention that Eq 2, 3 and 4 are complicated non-linear implicit equations in T_sfc as T_sfc also arise to the 4th power in the long wave upward component and in the expressions for H and LE.

5) P10 L21: integrated -> accumulated

6) P9 L6 When referring to Figure 2 the term S_soil has not been defined yet. As I understand it correctly, it is the left hand side of Eq 2 (with negative sign). Please clarify this in the text.

7) P10 L23. It is surprising that the reference scheme shows this instability. With a system with such large thermal inertia I would expect a stable solution. Can the authors comment on that?

8) P12 L1 extent -> magnitude

9) P16 L22 Why not mention approaches taken in other atmospheric models, like TESSEL in the ECMWF-model

---

## Referee Comment (RC2) · J. Ingwersen (Referee) · 17 May 2018

**General comments**

The authors investigated the effect of considering the canopy heat storage in closing the energy balance in the land surface model JSBACH. This is a very interesting and valuable approach forwards improving our current land surface models. In a first step the authors replaced the standard scheme of JSBACH for closing the energy balance at the land surface by a "diagnostic energy balance equation" (SkIn). The reader gets

the impression at this point that this is a novel approach. But it is not. This approach is used in many other land surface models (e.g. Noah-MP, Niu et al. 2011) for decades. The really novel and innovative aspect is that the authors consider the canopy heat storage in closing the energy balance at the land surface. At this point, unfortunately, the authors have missed that the heat capacity that they use in their model refers to the heat capacity of dry organic matter of biomass. Living plants, however, consist of 80% to 90% of water and the heat capacity of water is about 2.5 times higher than the one of organic matter. The correct approach is to use a weighted mean of both capacities (see Jacobs et al., 2008). Moreover, I have doubts that the calculation of the change of heat storage resulting from changes in canopy specific humidity is correct (see below). Therefore, I am afraid, the authors have to redo the SkIn+ simulations before this paper can be accepted.

Specific comments

I recommend to a add a paragraph to the Introduction about experimental studies on canopy heat storage (e.g. Jacobs et al., 2008; Meyers and Hollinger, 2004), so that the reader gets an idea of the magnitude of this storage term.

p. 3, line 2: At this point I wondered how the authors can study the coupling between the land and the atmosphere on the basis of offline simulations. Later on the authors state that JSBACH has a fully implicit land surface coupling scheme. I think it would be good to briefly introduce this scheme in more detail.

p. 3, line 29: If JSBACH computes the photosynthesis wouldn't it make sense to include also this flux in the energy balance equation? During the main growing period this flux is in a similar range or even higher than the canopy storage (see e.g. Jacobs et al., 2008; Meyers and Hollinger, 2004). This issue should be at least discussed.

p. 4, line 3-13: This part would better fit into the Introduction

p. 4, line 18: Please explain how the volumetric heat capacity is computed. As the

heat capacity is a function of the soil water content it is not constant in time. Therefore, I think it would be better to keep the capacity within the time derivative.

p. 5, line 14: Here it must be clearly stated that this is not a novel approach (see General comments).

p. 6, line 5: I would expect that the heat transfer coefficient is also a function of the soil water content as the soil water content affects the soil thermal diffusivity. Please discuss this issue.

p. 6, line 17: In my view, here something like a canopy porosity needs to be considered. Where biomass is, there is no air. In other words: within one cubic meter of canopy volume, the volume of air is less the one cubic meter. It is one cubic meter minus the volume of the biomass.

p. 6, line 21: This is not the equation that is used in Moore and Fisch (1986) for computing the heat storage change resulting from changes in specific humidity. I have doubts that this formula is correct. The specific humidity can also change at a constant surface temperature, e.g. due to a changing evapotranspiration as a response to a changing radiation. In Eq. 6 the capacity would be zero in such a situation as the derivative of qsat with respect to Tsfc is zero. Please describe in detail how you derived this equation and give the physical reasoning for this approach. Moreover, I think it would be better, instead of splitting the canopy heat capacity into three sub capacities, to split the canopy heat storage into three sub storage terms (heat storage change resulting from changes in canopy air temperature, specific humidity and biomass temperature (dry matter plus water)) as described in Moore and Fisch (1986) as well as in Jacobs et al. (2008). And please do not use the term "latent heat capacity of the air". Simply use the term "heat capacity". Otherwise it might be misleading.

p. 6, line 27: see General comments

p. 10, line 3-4: Eddy covariance measurements usually do not close the energy bal-

ance, i.e. the sum of the turbulent fluxes (latent and sensible heat flux) is smaller than the available energy (net radiation minus ground heat flux). The approach to compute the ground heat flux from the residuum of net radiation and latent and sensible heat flux implicates that the energy balance gap is entirely assigned to the ground heat flux. I am not aware of any other study that used such an approach. In most studies (see e.g., Twine et al., 2000; Ingwersen et al., 2015) it is assumed that the energy gap consists of latent and sensible heat and that the missing turbulent energy has the same Bowen ratio as the measured turbulent fluxes. This issue must be discussed!

p. 12, line 4: This wording is misleading. It sounds as the authors would consider twice the latent heat flux in the energy balance equation. This would be of course a severe mistake.

p. 15-16: The Conclusions must be streamlined and condensed. Many parts would better fit in the Discussion (e.g. p. 65, line 7-16).

Technical corrections

p. 1, line 11: Introduce the abbreviation AMIP.

p. 3, line 21: Delete "the model used in this study". That is clear at this point.

p. 8, line 26: Please introduce the abbreviation T63 resolution.

Figure 5: It would be better to plot both graphs over the same temperature range.

References

Ingwersen J., Imukova K., Högy P., Streck T., 2015. On the use of the post-closure methods uncertainty band to evaluate the performance of land surface models against eddy covariance flux data. Biogeosciences. 12, 2311-2326.

Meyers, T.P., Hollinger, S.E., 2004. An assessment of storage terms in the surface energy balance of maize and soybean. Agric. For. Meteorol. 125, 105–115. https://doi.org/10.1016/j.agrformet.2004.03.001

Niu G., Yang Z., Mitchell K.E., Chen F., Ek M.B., Barlage M., Kumar A., Manning K., Niyogi D., Rosero E., and others, 2011. The community Noah land surface model with multiparameterization options (Noah-MP): 1. Model description and evaluation with local-scale measurements. J. Geophys. Res. D: Atmospheres. 116.

Jacobs, A.F.G., Heusinkveld, B.G., Holtslag, A.A.M., 2008. Towards closing the surface energy budget of a mid-latitude grassland. Boundary-Layer Meteorol. 126, 125–136. https://doi.org/10.1007/s10546-007-9209-2

Twine, T.E., Kustas, W.P., Norman, J.M., Cook, D.R., Houser, P.R., Meyers, T.P., Prueger, J.H., Starks, P.J., Wesely, M.L., 2000. Correcting eddy-covariance flux underestimates over a grassland. Agric. For. Meteorol. 103 (3), 279-300.

———————————————————

---

## Referee Comment (RC3) · Anonymous Referee #3 · 22 May 2018

General Comment

The manuscript describes the application of the land component JSBACH of the MPI-ESM model on a short time scale as the diurnal cycle closing the energy balance for shallow vegetation within a soil layer of a finite heat capacity. Additionally, a new approach for the model in which the energy balance is closed within an infinitesimal thin surface soil layer is performed. Both approaches are compared with observations of net radiation, turbulent heat fluxes and ground heat flux obtained by eddy-covariance measurements during the CASES-99 experiment. Unfortunately, the ground heat flux

derived from the measurements is questionable (see Specific Comments below). The improvements of the new approach are stressed. Its impacts on the results of a global coupled land-atmosphere evaluation on a longer time scale are investigated in comparison to a run with the former approach. For this investigation the canopy heat capacity for regions with high vegetation is considered in JSBACH in a new scheme called SkIn+. The manuscript is well written and organized. The description of the models and of the changes made is understandable. The results are well discussed and the conclusions are comprehensible. The paper contributes to better represent weather and climate with models. Thus, I recommend the manuscript for publication in Geoscientific Model Development. However, I have some comments and questions to be answered before acceptance.

Specific Comments

P3, line 1-3: To my opinion, the question in bold letters relates to the reference model only and doesn't include the SkIn scheme. Or, is the SkIn scheme deemed to be correct?

P4, line 13: The term "offline experiment" first appears in the Introduction (line 32). Thus, the definition should already be given there.

P4, line 15: Up to which depth reaches the multi-layer vertical grid?

P6, Eq. (6): please define, whether the relative humidity within or above the canopy is meant.

P6, Eq. (6): Inserting Eq. (6) into Eq. (4) would lead to a time dependence of qsat which is more realistic than a simple dependence on Tsfc because qsat can also vary at constant temperature. Eq. (4) should be modified in this context.

P9, Fig.2 and P10, line 3,4: In contrast to DICE, in eddy-covariance experiments the ground heat flux is usually measured. Nevertheless, eddy-covariance generally doesn't close the energy balance. To close the balance, the missed energy (frequently exceeding 200 W m-2) is usually partitioned to the sensible and the latent heat flux according to the Bowen ratio. The full allocation of the residuum to the ground heat flux G leads to an overestimation of G which will be considerable for large residua. In turn, the green plots in Fig. 2 represent the sum of G and the residuum at daytime and G at nighttime when the residuum is close to zero. Please, comment this issue.

Technical Corrections

P7, Fig. 1: The yellow color is hardly visible. I suggest the authors should use another color for the incoming sw radiation.

---

## Author Comment (AC1) · 28 Jun 2018

Dear Referee #1, Please find enclosed a detailed version of our reply to your comments and the changes in the manuscript (including a difference file) in the attached supplement.

Please also note the supplement to this comment:
https://www.geosci-model-dev-discuss.net/gmd-2018-17/gmd-2018-17-AC1-supplement.zip

---

## Author Comment (AC2) · 28 Jun 2018

Dear Dr. Ingwersen, Please find enclosed a detailed version of our reply to your comments and the changes in the manuscript in the attached supplement.

Please also note the supplement to this comment: https://www.geosci-model-dev-discuss.net/gmd-2018-17/gmd-2018-17-AC2-supplement.zip

---

## Author Response (AR2)

Dear Referee #1,

before going into detail about your suggested improvements, we would like to thank you for taking the time to point out shortcomings and providing possible solutions for these. We feel that the proposed alterations, especially the more realistic representation of the latent heat storage increase the manuscript's quality significantly. In addition, we appreciate your impulse concerning the role of the skin conductivity in respect of an additional time phase shift. This encouraged us to get a better understanding which part of the energy balance has which effect in terms of dampening the system in time or in the magnitude of its amplitude.

**Main criticism**

1. *As one of the proposed changes in the surface model (SkIn) are based on ideas that has already been developed by others 20 years ago. See Viterbo and Beljaars, (1995) and the references to Betts et al. (1993) and Beljaars and Betts (1993), two points arise:*

   a) *It would be interesting to learn why these changes has not been introduced earlier in the JSBACH scheme. Were there, for example, other positive aspects in the performance of the JSBACH model that favored a conservative approach?*
   In the past, the focus of our Institute was on developing more complex representations of the water and carbon cycles on longer timescales. Therefore, the coupling of the atmosphere-canopy interface on short-time scales and in particular on the time step level was considered to be of minor importance, also because the effect of the soil heat storage was assumed to cancel out on average over long periods. An other aspect the existence of limited computational resources which prevented an iterative procedure to solve the energy balance equation which is however inevitable in the *SkIn* scheme.

   b) *It would be informative for the reader to compare the results/improvements found by the authors for the current model, with the findings of the authors mentioned above for the ECMWF-model.*
   Good idea, we compare the findings of Betts et al. (1993) with our results of the single-site experiment (page 13, line 6). However, the sole effect as the result of the introduction of a skin temperature (Viterbo and Beljaars, 1995) has not been tested in their study but rather the overall improvements due to their revised land surface scheme. Its effect for different regions on global scale remained unregarded, too.

2. *The rationale for Eq. 6 is unclear. A reference to Moore and Fisch (1986) is given, but I fail to see that their approach correspond to the approach given in the current paper. It is a change in q (specific humidity) that induces a change in latent heat storage in the vegetation air column. A change in q can occur while*

*$T_{\text{sfc}}$ stays constant. Thus Eq. 6 seems not to capture the process the authors try to describe.*

This is an issue that was addressed by all three referees and we agree that Eq. (6) is misleading without the derivation. The idea was to express the different types of canopy heat storages by means of heat capacities so that all heat storages could be related to the time derivative of the surface temperature. The reason behind this is that the surface temperature is the only prognostic variable to represent the processes in the canopy layer and the current scheme does not contain a prognostic variable like the specific humidity of the canopy air space. Thus, the heat storage resulting from changes in specific humidity in the canopy layer (in short: latent heat storage) $S_q$ was approximated by using the saturated values of specific humidity and the relative humidity within the canopy layer. In addition, we neglected the change of relative humidity within time ($\partial R_{\text{H}}/\partial t = 0$). So that $S_q$ can be written as follows:

$$
\begin{aligned}
S_q &= L_{\text{v}}\rho_{\text{a}}z_{\text{veg}}\frac{\partial q}{\partial t} = L_{\text{v}}\rho_{\text{a}}z_{\text{veg}}\frac{\partial R_{\text{H}}q_{\text{sat}}}{\partial t} \\
&= L_{\text{v}}\rho_{\text{a}}z_{\text{veg}}\left(R_{\text{H}}\frac{\partial q_{\text{sat}}}{\partial t} + q_{\text{sat}}\frac{\partial R_{\text{H}}}{\partial t}\right) \\
&\approx L_{\text{v}}\rho_{\text{a}}z_{\text{veg}}R_{\text{H}}\frac{\partial q_{\text{sat}}}{\partial t} \\
&\approx \underbrace{L_{\text{v}}\rho_{\text{a}}z_{\text{veg}}R_{\text{H}}\frac{\partial q_{\text{sat}}}{\partial T_{\text{sfc}}}}_{C_q}\frac{T_{\text{sfc}}}{\partial t}
\end{aligned}
\tag{1}
$$

where $q_{\text{sat}}$ is the saturated specific humidity at the surface temperature, $C_q$ the heat capacity related to humidity changes, $\rho_{\text{a}}$ the density of air, $z_{\text{veg}}$ the vegetation height and $L_{\text{v}}$ the latent heat of vaporization. We have to admit that the neglection of the time derivative of the relative humidity within the canopy layer is a rather crude approximation that may not be appropriate to estimate $S_q$.

As you have mentioned in your review, in using this approach we only consider changes in specific humidity due to changes in surface temperature and neglect other humidity sources and sinks. Therefore, we decided to develop an alternative parameterization for the latent heat storage which produces more realistic results for our purpose. We have addressed this issue in the manuscript, see from page 7, line 15 onwards. In this approach, we take into account the heat storage resulting from changes in specific humidity of the canopy air space by defining an effective *surface specific humidity* $q_{\text{sfc}}$ which is the best proxy for canopy specific humidity that we have. It represents a nonlinear weighted average between the specific air humidity above the canopy layer and the surface saturated specific humidity, by demanding that

$$
\frac{q_{\text{air}} - q_{\text{sfc}}}{r_{\text{a}}} \overset{!}{=} LE(q_{\text{air}}, q_{\text{sat}}, r_{\text{a}}, r_{\text{c}}, ...)
\tag{2}
$$

where $r_{\text{a}}$ is the atmospheric resistance, $r_{\text{c}}$ the canopy resistance and $LE$ the latent heat flux as it is calculated in the energy balance. This means that $q_{\text{sfc}}$ is

calculated to represent the effective near surface specific humidity that is required to reproduce the surface moisture fluxes due to turbulent exchange processes. In principle, the specific humidity of the boundary layer $q_{\mathrm{air}}$ could also be used as suggested by Moore and Fisch (1986). However, we are of the opinion that the usage of $q_{\mathrm{air}}$ would underestimate the latent heat storage in the current scheme. This leads to the new formulation of the latent heat storage $S_q$:

$$S_q = L_{\mathrm{v}} \rho_{\mathrm{a}} z_{\mathrm{veg}} \frac{\partial q_{\mathrm{sfc}}}{\partial t} \tag{3}$$

Because $q_{\mathrm{sfc}}$ is not a prognostic variable in the energy balance, its time derivative is approximated by using values of $q_{\mathrm{sfc}}$ at previous time steps. This is an approximation that is inevitable in the current model framework and can only be avoided by developing an extended dual source canopy layer scheme which includes a prognostic specific humidity of the canopy air space as mentioned in the discussion (chapter 4 of the manuscript).

Due to these changes in the parameterization of $S_q$, it is not possible anymore to compare the heat capacities related to different processes, but one has to compare heat storages (see chapter 3.2 of the manuscript). Because heat storages have the nature to compensate each other over longer time scales, we compare only positive contributions of the heat storages to estimate their magnitude. This could be interpreted as the average amount of energy that is stored in the canopy. The same amount will also be released.

Comparing the old approach of the latent heat storage (Eq. 1) with the new one (Eq. 3) on diurnal scales, we find that the old one tends to react like a common heat storage with a positive peak during the first half of the day and a negative during the second part (compare to the soil heat storage from Figure 2 of the manuscript). In contrast, the new representation of the latent heat storage does not exhibit this pattern. It shows positive as well as negative changes in heat storage during the whole daytime. This corresponds to the fact, that the specific humidity does not follow a strict diurnal pattern as the surface temperature. On the contrary, there are different kind of days representing either a positive or negative trend in humidity depending on wet or dry weather periods. The global mean over thirty years of the new representation of the latent heat storage is of the same magnitude as the old one. It reacts in slightly smaller values because the old one overestimated $S_q$ due to the direct coupling to the surface temperature.

3. *P11 L5-15: In principle I can follow the reasoning of the authors here, but I think the situation is a bit more complicated. Indeed $T_{\mathrm{sfc}}$ in the SkIn scheme responds instantaneously to the radiative forcing, but the coupling to the soil through the skin conductivity is also present. This may also induce time (phase) shifts. Please comment.*

You are right, at first view one could think that the skin conductivity should induce a phase shift, too. However, using a simplified model of this process it can

be illustrated that the skin conductivity like the drag coefficient – in contrast to the heat capacity – acts to reduce the relaxation time to reach the equilibrium. However, we agree that the incorporation of a skin conductivity also damps the amplitude of the response in surface temperature to variations in the forcing (see also page 13, line 17-19).

**Minor issues**

4. *For clarity it is good to mention that Eq 2, 3 and 4 are complicated non-linear implicit equations in $T_{\text{sfc}}$ as $T_{\text{sfc}}$ also arise to the 4th power in the long wave upward component and in the expressions for H and LE.*
   Good idea! Added (page 5, line 19-22) $\checkmark$

5. *P10 L21 integrated $\rightarrow$ accumulated*
   Changed (page 12, line 10) $\checkmark$

6. *P9 L6 When referring to Figure 2 the term $S_{\text{soil}}$ has not been defined yet. As I understand it correctly, it is the left hand side of Eq 2 (with negative sign). Please clarify this in the text.*
   Added (page 5, line 7) $\checkmark$

7. *P10 L23. It is surprising that the reference scheme shows this instability. With a system with such large thermal inertia I would expect a stable solution. Can the authors comment on that?*
   There are almost invisible, minor fluctuations in the surface temperature at time step level resulting from the numerical solution of the energy balance equation using the implicit numerical time stepping scheme, which can occur despite the large thermal inertia of the system. When plotting the soil heat storage, these variations become clearly visible due to the multiplication with the large value of the soil heat capacity of about $150\,000$ J/(m$^2$K).

8. *P12 L1 extent $\rightarrow$ magnitude*
   Changed (page 15, line 13) $\checkmark$

9. *P16 L22 Why not mention approaches taken in other atmospheric models, like TESSEL in the ECMWF-model*
   We addressed that by writing (page, line): *A more promising approach that would be more suitable for the SkIn$^+$ scheme and that allows a better representation of spatial subgrid-scale heterogeneity would be a flux aggregation method (Best et al., 2004; de Vrese and Hagemann, 2016) as it is used for example in the Tiled ECMWF Scheme for Surface Exchanges over Land model (TESSEL, Balsamo et al., 2009)*

Dear Dr. Ingwersen,

we would like to sincerely thank you for taking a lot of time to help us to improve our manuscript. It is immediately noticeable that you know the scientific field of our study very well. We feel that the large number of constructive criticisms and the requested alterations, most of all the different representation of the latent heat and biomass storage (the latter now containing the effect of moisture) as well as the introduction of a chemical heat storage, increase the manuscript's quality greatly. In addition, we appreciate your comment on the unclosure of the energy balance using eddy covariance measurements which removes some confusions concerning the calculation of the ground heat flux as a residuum term.

**Specific Comments**

- I recommend toan add a paragraph to the Introduction about experimental studies on canopy heat storage (e.g. Jacobs et al., 2008; Meyers and Hollinger, 2004), so that the reader gets an idea of the magnitude of this storage term.
  Good Idea! Added (page 3, line 23-27)

- p. 3, line 2: At this point I wondered how the authors can study the coupling between the land and the atmosphere on the basis of offline simulations.
  We clarified this issue by rephrasing the scientific question (page 3, line 14-16).
  Later on the authors state that JSBACH has a fully implicit land surface coupling scheme. I think it would be good to briefly introduce this scheme in more detail.
  Added (page 4, line 10-12) $\checkmark$

- p. 3, line 29: If JSBACH computes the photosynthesis wouldn't it make sense to include also this flux in the energy balance equation? During the main growing period this flux is in a similar range or even higher than the canopy storage (see e.g. Jacobs et al., 2008; Meyers and Hollinger, 2004). This issue should be at least discussed.
  Good advice! The energy balance equation now includes the energy of photosynthesis (see from page 8, line 9 onwards).

- p. 4, line 3-13: This part would better fit into the Introduction
  Yes, that makes more sense! Changed (from page 2, line 28 onwards) $\checkmark$

- p. 4, line 18: Please explain how the volumetric heat capacity is computed. As the heat capacity is a function of the soil water content it is not constant in time. Therefore, I think it would be better to keep the capacity within the time derivative.
  The volumetric heat capacity originates from FAO maps and does not contain a dependence on the soil moisture.

- p. 5, line 14: Here it must be clearly stated that this is not a novel approach (see General comments).

We have addressed this issue from page 5 line 31 onwards by writing: *Of course, we have to admit that the use of the instantaneous response temperature is not a novel approach. This so-called skin temperature has been introduced by Viterbo and Beljaars (1995) to replace the old ground-surface model of the ECMWF. This approach is also used in other land surface models, e.g. in the community Noah land surface model (Niu et al., 2011).*

- p. 6, line 5: I would expect that the heat transfer coefficient is also a function of the soil water content as the soil water content affects the soil thermal diffusivity. Please discuss this issue.

The heat transfer coefficient is an empirical quantity which describes the thermal connection between the soil and the surface. For tall vegetation this means for example that it mainly estimates the thermal transport within the canopy that is not known unless the turbulence in the canopy layer is approximated, e.g. in a more complex canopy layer scheme. Regarding this, we think that the soil water is of secondary importance for the heat transfer coefficient and further experimental investigations would be needed. However, this would go beyond the scope of this work.

- p. 6, line 17: In my view, here something like a canopy porosity needs to be considered. Where biomass is, there is no air. In other words: within one cubic meter of canopy volume, the volume of air is less the one cubic meter. It is one cubic meter minus the volume of the biomass.

You are right, but we estimated this factor using values given by Moore and Fisch (1986) and have concluded that this factor is definitely smaller than 1 %. This is the reason why we are neglecting it.

- p. 6, line 21: This is not the equation that is used in Moore and Fisch (1986) for computing the heat storage change resulting from changes in specific humidity. I have doubts that this formula is correct. The specific humidity can also change at a constant surface temperature, e.g. due to a changing evapotranspiration as a response to a changing radiation. In Eq. 6 the capacity would be zero in such a situation as the derivative of qsat with respect to Tsfc is zero. Please describe in detail how you derived this equation and give the physical reasoning for this approach. Moreover, I think it would be better, instead of splitting the canopy heat capacity into three sub capacities, to split the canopy heat storage into three sub storage terms (heat storage change resulting from changes in canopy air temperature, specific humidity and biomass temperature (dry matter plus water)) as described in Moore and Fisch (1986) as well as in Jacobs et al. (2008). And please do not use the term "latent heat capacity of the air". Simply use the term "heat capacity". Otherwise it might be misleading.

This is an issue that was addressed by all three referees and we agree that Eq. (6) is misleading without the derivation. The idea was to express the different types of canopy heat storages by means of heat capacities so that all heat storages could be related to the time derivative of the surface temperature. The reason behind

this is that the surface temperature is the only prognostic variable to represent the processes in the canopy layer and the current scheme does not contain a prognostic variable like the specific humidity of the canopy air space. Thus, the heat storage resulting from changes in specific humidity in the canopy layer (in short: latent heat storage) $S_q$ was approximated by using the saturated values of specific humidity and the relative humidity within the canopy layer. In addition, we neglected the change of relative humidity within time ($\partial R_H / \partial t = 0$). So that $S_q$ can be written as follows:

$$
\begin{aligned}
S_q = L_v \rho_a z_{veg} \frac{\partial q}{\partial t} &= L_v \rho_a z_{veg} \frac{\partial R_H q_{sat}}{\partial t} \\
&= L_v \rho_a z_{veg} \left( R_H \frac{\partial q_{sat}}{\partial t} + q_{sat} \frac{\partial R_H}{\partial t} \right) \\
&\approx L_v \rho_a z_{veg} R_H \frac{\partial q_{sat}}{\partial t} \\
&\approx \underbrace{L_v \rho_a z_{veg} R_H \frac{\partial q_{sat}}{\partial T_{sfc}}}_{C_q} \frac{T_{sfc}}{\partial t}
\end{aligned}
\tag{1}
$$

where $q_{sat}$ is the saturated specific humidity at the surface temperature, $C_q$ the heat capacity related to humidity changes, $\rho_a$ the density of air, $z_{veg}$ the vegetation height and $L_v$ the latent heat of vaporization. We have to admit that the neglection of the time derivative of the relative humidity within the canopy layer is a rather crude approximation that may not be appropriate to estimate $S_q$.

As you have mentioned in your review, in using this approach we only consider changes in specific humidity due to changes in surface temperature and neglect other humidity sources and sinks. Therefore, we decided to develop an alternative parameterization for the latent heat storage which produces more realistic results for our purpose. We have addressed this issue in the manuscript, see from page 7, line 15 onwards. In this approach, we take into account the heat storage resulting from changes in specific humidity of the canopy air space by defining an effective *surface specific humidity* $q_{sfc}$ which is the best proxy for canopy specific humidity that we have. It represents a nonlinear weighted average between the specific air humidity above the canopy layer and the surface saturated specific humidity, by demanding that

$$
\frac{q_{air} - q_{sfc}}{r_a} \stackrel{!}{=} LE(q_{air}, q_{sat}, r_a, r_c, ...)
\tag{2}
$$

where $r_a$ is the atmospheric resistance, $r_c$ the canopy resistance and $LE$ the latent heat flux as it is calculated in the energy balance. This means that $q_{sfc}$ is calculated to represent the effective near surface specific humidity that is required to reproduce the surface moisture fluxes due to turbulent exchange processes. In principle, the specific humidity of the boundary layer $q_{air}$ could also be used as suggested by Moore and Fisch (1986). However, we are of the opinion that the usage of $q_{air}$ would underestimate the latent heat storage in the current scheme.

This leads to the new formulation of the latent heat storage $S_q$:

$$S_q = L_v \rho_a z_{veg} \frac{\partial q_{sfc}}{\partial t} \quad (3)$$

Because $q_{sfc}$ is not a prognostic variable in the energy balance, its time derivative is approximated by using values of $q_{sfc}$ at previous time steps. This is an approximation that is inevitable in the current model framework and can only be avoided by developing an extended dual source canopy layer scheme which includes a prognostic specific humidity of the canopy air space as mentioned in the discussion (chapter 4 of the manuscript).

Due to these changes in the parameterization of $S_q$, it is not possible anymore to compare different heat capacities, but one has to compare heat storages of different processes (see chapter 3.2 of the manuscript). Because heat storages have the nature to compensate each other over longer time scales, we compare only positive contributions of the heat storages to estimate their magnitude. This could be interpreted as the average amount of energy that is stored in the canopy and the same amount will also be released.

Comparing the old approach of the latent heat storage (Eq. 1) with the new one (Eq. 3) on diurnal scales, we find that the old one tends to react like a common heat storage with a positive peak during the first half of the day and a negative during the second part (compare to the soil heat storage from Figure 2 of the manuscript). In contrast, the new representation of the latent heat storage does not exhibit this pattern. It shows positive as well as negative changes in heat storage during the whole daytime. This corresponds to the fact, that the specific humidity does not follow a strict diurnal pattern as the surface temperature. On the contrary, there are different kind of days representing either a positive or negative trend in humidity depending on wet or dry weather periods. The global mean over thirty years of the new representation of the latent heat storage is of the same magnitude as the old one. It reacts in slightly smaller values because the old one overestimated $S_q$ due to the direct coupling to the surface temperature.

- p. 6, line 27: see General comments (At this point, unfortunately, the authors have missed that the heat capacity that they use in their model refers to the heat capacity of dry organic matter of biomass. Living plants, however, consist of 80 % to 90 % of water and the heat capacity of water is about 2.5 times higher than the one of organic matter. The correct approach is to use a weighted mean of both capacities (see Jacobs et al., 2008).)
  This is a crucial aspect and we are glad you made it up. We introduced the heat storage of moist biomass on page 7 from line 4 onwards and discussed its effect in chapter 3.2 of the manuscript.

- p. 10, line 3-4: Eddy covariance measurements usually do not close the energy balance, i.e. the sum of the turbulent fluxes (latent and sensible heat flux) is smaller than the available energy (net radiation minus ground heat flux). The

approach to compute the ground heat flux from the residuum of net radiation and latent and sensible heat flux implicates that the energy balance gap is entirely assigned to the ground heat flux. I am not aware of any other study that used such an approach. In most studies (see e.g., Twine et al., 2000; Ingwersen et al., 2015) it is assumed that the energy gap consists of latent and sensible heat and that the missing turbulent energy has the same Bowen ratio as the measured turbulent fluxes. This issue must be discussed!

We do totally agree and are aware of the unclosure of the energy balance using the eddy covariance method. However, the problem is, if the ground heat flux was not measured, there is no possibility to estimate the imbalance and therefore to divide it into sensible and latent heat flux part. Thus, in our opinion, it makes more sense to depict the ground heat flux including a possible imbalance than discarding it completely. Nonetheless, you are right that we should at least mention the imbalance to avoid confusions.

- p. 12, line 4: This wording is misleading. It sounds as the authors would consider twice the latent heat flux in the energy balance equation. This would be of course a severe mistake.
  You are right! We changed the whole paragraph due to the modifications for the biomass and latent heat storage (see chapter 3.2 of the manuscript) and avoided this misleading formulation.

- p. 15-16: The Conclusions must be streamlined and condensed. Many parts would better fit in the Discussion (e.g. p. 65, line 7-16).
  Fair point! Changed (see chapter 4 and 5 of the manuscript) $\checkmark$

**Technical corrections**

- p. 1, line 11: Introduce the abbreviation AMIP.
  Added (page 1, line 11) $\checkmark$

- p. 3, line 21: Delete "the model used in this study". That is clear at this point.
  Removed at this point and added in the new part (based on your above mentioned suggestion) of the introduction where it is more suitable (page 2, line 28) $\checkmark$

- p. 8, line 26: Please introduce the abbreviation T63 resolution.
  Added (page 10, line 15) $\checkmark$

- Figure 5: It would be better to plot both graphs over the same temperature range.
  Changed $\checkmark$

Dear Referee #3,

we would like to thank you for taking your time to help us to improve our manuscript. We think that your suggested comments, in particular the alternative representation of the latent heat storage (not only including a simple dependence on the surface temperature), increase the manuscript's quality significantly. Moreover, we appreciate your comment on the ground heat flux which removes some confusions concerning the unclosure of the energy balance using eddy covariance measurements.

**Specific Comments**

- P3, line 1-3: To my opinion, the question in bold letters relates to the reference model only and doesn't include the SkIn scheme. Or, is the SkIn scheme deemed to be correct?
  Good point! We clarified this issue by rephrasing the scientific question (page 3, line 14-16): *Does the SkIn scheme improve the performance in reproducing the diurnal cycle in comparison to the old heat storage concept in case of shallow vegetation?*

- P4, line 13: The term "offline experiment" first appears in the Introduction (line 32). Thus, the definition should already be given there.
  Changed (page 3, line 10-11) $\checkmark$

- P4, line 15: Up to which depth reaches the multi-layer vertical grid?
  It reaches up to a depth of 10 m (page 4, line 22).

- P6, Eq. (6): Please define, whether the relative humidity within or above the canopy is meant.
  That was one problem of the old latent heat storage. We estimated the relative humidity within the canopy as follows:

$$R_{\mathrm{H}} = \frac{q_{\mathrm{air}}}{q_{\mathrm{sat}}(T_{\mathrm{sfc}})} \tag{1}$$

This is clearly not correct and not needed anymore in the new formulation of the latent heat storage (see next comment).

- P6, Eq. (6): Inserting Eq. (6) into Eq. (4) would lead to a time dependence of qsat which is more realistic than a simple dependence on Tsfc because qsat can also vary at constant temperature. Eq. (4) should be modified in this context.
  This is an issue that was addressed by all three referees and we agree that Eq. (6) is misleading without the derivation. The idea was to express the different types of canopy heat storages by means of heat capacities so that all heat storages could be related to the time derivative of the surface temperature. The reason behind this is that the surface temperature is the only prognostic variable to represent the processes in the canopy layer and the current scheme does not contain a prognostic

variable like the specific humidity of the canopy air space. Thus, the heat storage resulting from changes in specific humidity in the canopy layer (in short: latent heat storage) $S_q$ was approximated by using the saturated values of specific humidity and the relative humidity within the canopy layer. In addition, we neglected the change of relative humidity within time ($\partial R_{\mathrm{H}}/\partial t = 0$). So that $S_q$ can be written as follows:

$$
\begin{aligned}
S_q &= L_{\mathrm{v}}\rho_{\mathrm{a}}z_{\mathrm{veg}}\frac{\partial q}{\partial t} = L_{\mathrm{v}}\rho_{\mathrm{a}}z_{\mathrm{veg}}\frac{\partial R_{\mathrm{H}}q_{\mathrm{sat}}}{\partial t} \\
&= L_{\mathrm{v}}\rho_{\mathrm{a}}z_{\mathrm{veg}}\left(R_{\mathrm{H}}\frac{\partial q_{\mathrm{sat}}}{\partial t} + q_{\mathrm{sat}}\frac{\partial R_{\mathrm{H}}}{\partial t}\right) \\
&\approx L_{\mathrm{v}}\rho_{\mathrm{a}}z_{\mathrm{veg}}R_{\mathrm{H}}\frac{\partial q_{\mathrm{sat}}}{\partial t} \\
&\approx \underbrace{L_{\mathrm{v}}\rho_{\mathrm{a}}z_{\mathrm{veg}}R_{\mathrm{H}}\frac{\partial q_{\mathrm{sat}}}{\partial T_{\mathrm{sfc}}}}_{C_q}\frac{T_{\mathrm{sfc}}}{\partial t}
\end{aligned}
\tag{2}
$$

where $q_{\mathrm{sat}}$ is the saturated specific humidity at the surface temperature, $C_q$ the heat capacity related to humidity changes, $\rho_{\mathrm{a}}$ the density of air, $z_{\mathrm{veg}}$ the vegetation height and $L_{\mathrm{v}}$ the latent heat of vaporization. We have to admit that the neglection of the time derivative of the relative humidity within the canopy layer is a rather crude approximation that may not be appropriate to estimate $S_q$.

As you have mentioned in your review, in using this approach we only consider changes in specific humidity due to changes in surface temperature and neglect other humidity sources and sinks. Therefore, we decided to develop an alternative parameterization for the latent heat storage which produces more realistic results for our purpose. We have addressed this issue in the manuscript, see from page 7, line 15 onwards. In this approach, we take into account the heat storage resulting from changes in specific humidity of the canopy air space by defining an effective *surface specific humidity* $q_{\mathrm{sfc}}$ which is the best proxy for canopy specific humidity that we have. It represents a nonlinear weighted average between the specific air humidity above the canopy layer and the surface saturated specific humidity, by demanding that

$$
\frac{q_{\mathrm{air}} - q_{\mathrm{sfc}}}{r_{\mathrm{a}}} \stackrel{!}{=} LE(q_{\mathrm{air}}, q_{\mathrm{sat}}, r_{\mathrm{a}}, r_{\mathrm{c}}, ...)
\tag{3}
$$

where $r_{\mathrm{a}}$ is the atmospheric resistance, $r_{\mathrm{c}}$ the canopy resistance and $LE$ the latent heat flux as it is calculated in the energy balance. This means that $q_{\mathrm{sfc}}$ is calculated to represent the effective near surface specific humidity that is required to reproduce the surface moisture fluxes due to turbulent exchange processes. In principle, the specific humidity of the boundary layer $q_{\mathrm{air}}$ could also be used as suggested by Moore and Fisch (1986). However, we are of the opinion that the usage of $q_{\mathrm{air}}$ would underestimate the latent heat storage in the current scheme. This leads to the new formulation of the latent heat storage $S_q$:

$$
S_q = L_{\mathrm{v}}\rho_{\mathrm{a}}z_{\mathrm{veg}}\frac{\partial q_{\mathrm{sfc}}}{\partial t}
\tag{4}
$$

Because $q_{\mathrm{sfc}}$ is not a prognostic variable in the energy balance, its time derivative is approximated by using values of $q_{\mathrm{sfc}}$ at previous time steps. This is an approximation that is inevitable in the current model framework and can only be avoided by developing an extended dual source canopy layer scheme which includes a prognostic specific humidity of the canopy air space as mentioned in the discussion (chapter 4 of the manuscript).

Due to these changes in the parameterization of $S_q$, it is not possible anymore to compare different heat capacities, but one has to compare heat storages of different processes (see chapter 3.2 of the manuscript). Because heat storages have the nature to compensate each other over longer time scales, we compare only positive contributions of the heat storages to estimate their magnitude. This could be interpreted as the average amount of energy that is stored in the canopy and the same amount will also be released.

Comparing the old approach of the latent heat storage (Eq. 2) with the new one (Eq. 4) on diurnal scales, we find that the old one tends to react like a common heat storage with a positive peak during the first half of the day and a negative during the second part (compare to the soil heat storage from Figure 2 of the manuscript). In contrast, the new representation of the latent heat storage does not exhibit this pattern. It shows positive as well as negative changes in heat storage during the whole daytime. This corresponds to the fact, that the specific humidity does not follow a strict diurnal pattern as the surface temperature. On the contrary, there are different kind of days representing either a positive or negative trend in humidity depending on wet or dry weather periods. The global mean over thirty years of the new representation of the latent heat storage is of the same magnitude as the old one. It reacts in slightly smaller values because the old one overestimated $S_q$ due to the direct coupling to the surface temperature.

- P9, Fig. 2 and P10, line 3,4: In contrast to DICE, in eddy-covariance experiments the ground heat flux is usually measured. Nevertheless, eddy-covariance generally doesn't close the energy balance. To close the balance, the missed energy (frequently exceeding 200 W/m$^2$) is usually partitioned to the sensible and the latent heat flux according to the Bowen ratio. The full allocation of the residuum to the ground heat flux $G$ leads to an overestimation of $G$ which will be considerable for large residua. In turn, the green plots in Fig. 2 represent the sum of $G$ and the residuum at daytime and $G$ at nighttime when the residuum is close to zero. Please, comment this issue.

We do totally agree and are aware of the unclosure of the energy balance using the eddy covariance method. However, the problem is, if the ground heat flux was not measured, there is no possibility to estimate the imbalance and therefore to divide it into sensible and latent heat flux part. Thus, in our opinion, it makes more sense to depict the ground heat flux including a possible imbalance than discarding it completely. Nonetheless, you are right that we should at least mention the imbalance to avoid confusions.

**Technical Corrections**

- P7, Fig. 1: The yellow color is hardly visible. I suggest the authors should use another color for the incoming sw radiation.
  Changed $\sqrt{}$

[revised manuscript text omitted]

$$\sout{C_T}S_{\mathrm{cano}} = \sout{c_p\rho_a z}\,S_T + S_{\mathrm{veg}\_} \sout{+S_q} \tag{5}$$

where $S_T$ denotes the heat  storage in the canopy air space, $S_{\mathrm{veg}}$ the heat storage of biomass and $S_q$ the heat storage resulting from changes in specific humidity

20 in the canopy layer (in short latent heat storage). The heat storage in the canopy air space $S_T$ can be expressed as

$$S_T = C_T\frac{\partial T_{\mathrm{sfc}}}{\partial t} = c_p\rho_a z_{\mathrm{veg}}\frac{\partial T_{\mathrm{sfc}}}{\partial t} \tag{6}$$

where $C_T$ is the area-specific heat capacity of the canopy air, $c_p = 1005$ J/(kg K) the specific heat capacity of air at constant pressure. , $\rho_a$ the density of air and $z_{\mathrm{veg}}$ the vegetation height. The heat storage of biomass in the canopy layer $S_{\mathrm{veg}}$ is determined as

$$S_{\mathrm{veg}} = C_{\sout{q}\,\mathrm{veg}}\frac{\partial T_{\mathrm{sfc}}}{\partial t} = L_v\rho_a R_H z\,\sout{c_{\mathrm{veg}}}\frac{\partial q_{\mathrm{sat}}}{\partial T_{\mathrm{sfc}}}\sout{m_{\mathrm{veg}}}\frac{\partial T_{\mathrm{sfc}}}{\partial t} \tag{7}$$

where  $C_{\mathrm{veg}}$ is the area-specific heat capacity of the biomass, $m_{\mathrm{veg}}$ the area specific mass of biomass and $c_{\mathrm{veg}}$ the specific heat capacity of moist biomass according to Moore and Fisch (1986). The latter is approximated by a weighted average between the specific heat capacity of dry biomass containing a temperature dependence and the specific heat capacity of water $c_w = 4184$ J/(kg K) assuming a constant water mixing ratio. For example, at a temperature of 25 °C

the canopy biomass has a specific heat capacity of $c_{veg} \approx 2650$ J/(kg K). The area specific mass of moist biomass $m_{veg}$ can be estimated as a function of the vegetation height $z_{veg}$ using a linear relationship, namely $m_{veg} = \rho_{veg} z_{veg}$, where $\rho_{veg} \approx 1.67 \frac{kg}{m^3}$ is the partial density of moist biomass, i.e. the mass of moist biomass per one cubic meter of air estimated using values given by Moore and Fisch (1986).

The latent heat storage $S_q$ can be calculated according to Moore and Fisch (1986) as follows:

$$S_q = L_v \rho_a z_{veg} \frac{\partial q_{cas}}{\partial t} \tag{8}$$

$C_{veg} = c_{veg} m_{veg}$

 where $L_v = 2.5 \cdot 10^6$ J/kg denotes the latent heat of vaporization and $q_{cas}$ the specific humidity in the canopy  air space. In contrast to the heat  storages of canopy air and biomass – Eq. (6) and Eq. (7) – which are expressed by means of heat capacities related to the time derivative of surface temperature, the situation is more complicated regarding the latent heat storage: Changes in specific humidity can occur independently of temperature changes. That means, considering only changes in specific humidity due to changes in surface temperature would neglect other humidity sources and sinks. Thus, a different approach to parameterize the latent heat storage is required because the current schemes does not contain a prognostic variable for the specific humidity in the canopy air space. In this approach, we take into account the heat storage resulting from changes in specific humidity of the canopy air space by defining an effective *surface specific humidity* $q_{sfc}$, which is the best proxy for the specific humidity in the canopy layer that we have. It represents a nonlinear weighted average between the specific air humidity above the canopy $q_{air}$ and the saturated specific humidity at the surface temperature $q_{sat}(T_{sfc})$, by demanding that

[revised manuscript text omitted]